# How to mitigate flood events similar to the 1979 catastrophic floods in lower Tagus

Diego Fernández-Nóvoa[1,2], Alexandre M. Ramos[3], José González-Cao[1], Orlando García-Feal[1], Cristina Catita[2], Moncho Gómez-Gesteira[1], Ricardo M. Trigo[2,4]

[1]Centro de Investigación Mariña (CIM), Universidade de Vigo, Environmental Physics Laboratory (EPhysLab), Campus da Auga, 32004 Ourense, Spain
[2]Instituto Dom Luiz (IDL), Faculdade de Ciências da Universidade de Lisboa, 1749-016 Lisbon, Portugal
[3]Institute of Meteorology and Climate Research, Karlsruhe Institute of Technology, Karlsruhe, Germany
[4]Departamento de Meteorologia, Universidade Federal Do Rio de Janeiro, Rio de Janeiro, Brazil

*Correspondence to*: Diego Fernández-Nóvoa (diefernandez@uvigo.es)

**Abstract.** The floods that struck lower Tagus valley in February 1979 correspond to the most intense floods in this river and affected the largest number of people in a river flow event in Portugal, during the last 150 years. In fact, the vast area affected impacted significantly circa 10k people in the lower Tagus sector (and an additional 7k in other regions of Portugal), including thousands of people evacuated or made homeless. In this context, the present study focuses on an in depth analysis of this event from a hydrodynamic perspective by means of the Iber+ numerical model and on developing strategies to mitigate the flood episodes that occur in the lower section of the Tagus River using the outstating floods of February 1979 as benchmark. In this sense, dam operating strategies were developed and analyzed for the most important dam along the Tagus River basin in order to propose effective procedures to take advantage of these infrastructures to minimize the effect of floods. Overall, the numerical results indicate a good agreement with water marks and some descriptions of the 1979 flood event, which demonstrates the model capability to evaluate floods in the area under study. Regarding flood mitigation, obtained results indicate that the frequency of floods can be reduced with the proposed strategies, which were focused on providing optimal dam operating rules to mitigate flooding in lower Tagus valley. In addition, hydraulic simulations corroborated an important decrease in water depth and velocity for the most extreme flood events, and also a certain reduction of flood extension was detected. This confirms the effectiveness of the proposed strategies to help in reducing flood impact in lower Tagus valley through the efficient functioning of dams.

# 1 Introduction

The Iberian Peninsula corresponds to a relevant region that has been historically affected by intense river floods (Benito et al., 1996; 2003; Pereira et al., 2016; Rebelo et al., 2018; Santos et al., 2018; González-Cao et al., 2021; 2022). In particular, its western area is especially vulnerable to these phenomena since it can be directly affected by the storm-tracks of the Northern Hemisphere that transport heat and moisture (Peixoto and Oort, 1992; Trigo, 2006). Some specific synoptic features can favor extreme weather conditions that promote high precipitation rates and thus important associated river floods (Trigo and DaCamara, 2000; Trigo, 2006; Rebelo et al., 2018). In this context, in the last decade there has been a steep increment of works dealing with historical river floods that took place in the western half of the Iberian Peninsula, namely the floods that occurred in the Minho, Lima and Douro basins in 1909 (Pereira et al., 2016), in the Tagus basin in 1876 and 1979 (Benito et al., 2003; Salgueiro et al., 2013; Trigo et al., 2014; Rebelo et al., 2018), or in the Guadiana basin in 1876 (Trigo et al., 2014; González-Cao et al., 2021). The floods that struck lower Tagus valley from February 5 to 16, 1979 correspond to the most intense occurred in this river since, at least, the mid-19$^{th}$ century. Additionally, this outstanding event also implied to the largest number of people affected in the Iberian Peninsula in a river flood event in the last 150 years. The area affected by prolonged precipitation is much larger, thus other regions of western Iberia were also seriously affected. In fact, the vast area affected impacted roughly 10k people in the lower Tagus sector (and an additional 7k in other regions of Portugal), including thousands of people evacuated or made homeless. A detailed description of the causes behind this event and its main consequences can be found in Rebelo et al. (2018). Unlike previous more historical events (such as the 1876 floods), in 1979 there were already several large dams and reservoirs in both the Spanish and Portuguese sections of the Tagus River basin, which could have mitigated the amplitude of the flood. These structures are one of the main mechanisms for flood reduction (Lee et al., 2009; Valeriano et al., 2010; Chou and Wu, 2015), indeed, certain research endeavors have been undertaken to identify optimal sites for the establishment of new dams, aiming to tackle this problem, among other objectives (Pathan et al., 2022). However, lack of communications between the management authorities in both countries at the time (a joint action protocol was not established until the beginning of the 21$^{st}$ century; https://www.boe.es/diario_boe/txt.php?id=BOE-A-2000-2882 and Escartín, 2002), coupled with poor dam operations, was translated in leading to bad performance in terms of mitigating the 1979 flood (Rebelo et al., 2018). When flow peaks arrived, dams controlling Tagus flow were close to full capacity therefore hampering the ability to exert sufficient control on the peak river flow (Rebelo et al., 2018). Thus, the main aims of the present study are: i) to reproduce and analyze this historical event from a hydrologic-hydraulic point of view using the Iber+ numerical model; ii) the development of dam operating strategies to mitigate the flood episodes that occur in the lower section of the Tagus River using the outstating floods of February 1979 as benchmark. The effectiveness of the strategies proposed in terms of reducing the flood impact will be evaluated by means of numerical model simulations. In this sense, hydrodynamic numerical models are a key tool in flood risk management by providing information on critical flood indicators (Shah et al., 2022; Zhang et al., 2022). This will contribute to address the flood mitigation challenges that the scientific community will face in the coming decades (IPCC,

2012; 2021; Prieto et al., 2020). These are related to the increasing temperatures due to the significant rates of global warming that contributes to the recent and future increase of extreme precipitation and floods in some parts of the world (Dankers and Feyen, 2008; Petrow and Merz, 2009; Alfieri et al., 2015; 2017; Diakakis, 2016; Arnell and Gosling, 2016; Modarres et al., 2016; Jongman, 2018; IPCC 2021), including vast areas of the Iberian Peninsula (Lorenzo and Alvarez, 2020). Additionally, this study also intends to prove that with open data and free applications for modeling, the results are

satisfactory enough to apply the methodology proposed here in other regions where more detailed data may be scarce or non-existent.

This document is organized as follows: in Section 2 a brief motivation of the study is provided. In Section 3 a brief description of the area under scope is presented. In Section 4, the data required to develop the study, the hydrodynamic model used, and a proposal of dam operating strategies to mitigate lower Tagus floods, are described. Section 5 is devoted to

results, including: i) testing different Digital Elevation Models (DEMs) to select the most accurate for the area under scope, ii) the modelling and in-depth analysis of the flood registered in 1979 in lower Tagus valley, iii) the analysis of the effectiveness of proposed dam operating strategies to mitigate floods in lower Tagus valley. Finally, the main conclusions are summarized in section 6.

## 2 Motivation

The main motivations driving this study are, on the one hand, to improve the knowledge and understanding of flood development in lower Tagus valley, an area especially vulnerable to these events. In this sense, one of the main motivations for carrying out this analysis was the scarcity of studies available addressing this issue, especially from a hydrodynamic point of view. For that, different freely available products were tested in order to provide the most accurate tools that can serve as a basis for future studies focused on addressing different aspects related to flooding in lower Tagus valley. On the

other hand, the study also intends to provide different strategies to mitigate floods in lower Tagus valley but taking advantage of existing infrastructures, namely large dams. To the best of our knowledge, there are no previous studies that have developed this type of strategies for the area under scope, so the strategies presented in this work could represent an important advance in this field. This proposal will allow to provide an affordable new approach to flood mitigation compared to the implementation of additional structural measures that have to be built. For that, dam operating strategies will be

proposed and tested in the most important Tagus dam. The benefits provided by the dam strategies proposed in relation to flood mitigation, will be also evaluated. This will also serve as a basis for developing future studies focused on optimizing dam strategies or even interconnecting the strategies of different dams of the Tagus basin to improve the flood mitigation.

## 3 Area of study

The international Tagus basin is one of the largest river basins located in the Iberian Peninsula, draining more than 80,000 km$^2$ (approximately 70% in Spain and 30% in Portugal) (Ramos and Reis, 2001). The Tagus River flows from an elevation close to 1600 m.a.s.l. in the headwaters (Sierra de Albarracín, Spain) to sea level at its mouth in the Atlantic Ocean in Lisbon, with an approximate length of 1100 km (Agência Portuguesa do Ambiente, 2016) (Figure 1). Its flow is characterized by a pluvial regime with higher discharges in winter months and lower ones in summer, with a mean discharge of approximately 450 m$^3$s$^{-1}$ (Ramos and Reis, 2001; Fernández-Nóvoa et al., 2017). The natural regime of the Tagus River was highly modified following the construction of several reservoirs in the main river but also in some of its tributaries. Among them, the Alcántara dam stands out by its sheer volume, located on the border between Spain and Portugal (Figure 1), with a total capacity surpassing 3160 hm$^3$, being the second most important reservoir in the Iberian Peninsula and presenting a high capacity for flow retention. It drains a total extent larger than 50,000 km$^2$, which supposes about 70% of the total extent of Tagus basin. The capacity and location of Alcántara dam imply that the Tagus River flow in the Portuguese sector is controlled, to a large extent, by this dam (Rebelo et al., 2018). Therefore, operating strategies focused on mitigating floods in the lower Tagus valley will be developed and applied to the dam of Alcántara as a case study.

Finally, it is important to remark that the particular characteristics of the lower Tagus valley are especially relevant for the scope of this study. This valley is characterized by a Cenozoic sedimentary basin, with a large and flattened alluvial plain, which promotes that floods can affect large extensions causing important damages (Rebelo et al., 2018) (Figure 1). This makes this area highly susceptible to periodic flooding associated with different phenomena, such as upstream hydrological flood, downstream tidal floods or storm surge (Vargas et al., 2008; Rocha et al., 2020; Lopes et al., 2022).

## 4 Data, Models and Methods

### 4.1 Data

Precipitation data for the area under scope were obtained from Iberia01 database (Herrera et al., 2019). This dataset provides daily precipitation data with the highest spatial resolution of 0.1° (≈ 10 km) covering the entire Iberian Peninsula available at the moment. It was produced using a dense network of stations spanning the period 1971-2015. Data are freely available on  https://doi.org/10.20350/digitalCSIC/8641.

Daily mean Tagus River discharge data at the Almourol station (Figure 1) were downloaded from the SNIRH (Sistema Nacional de Informação de Recursos Hídricos) database for the period 1973-2021. The SNIRH is the National Information System on Water Resources of Portugal, whose data are freely available on www.snirh.pt.

Information of the level reached by the water in different points of the lower Tagus valley during the flood event that occurred in February 1979, were also obtained from the SNIRH database (see control points (black circles) in Figure 1).

Daily outflow and volume data at the Alcántara dam were provided by CEDEX (Centro de Estudios y Experimentación de Obras Públicas) for the period from October 1970 to September 2019. CEDEX is the Spanish institution in charge of managing part of the hydrologic data of the country, which are freely available on http://www.cedex.es/.

## 4.2 Hydraulic model

Iber is a numerical model that solves the 2D depth-averaged shallow water equations applying the finite volume methodology (Bladé et al., 2014). Recently, this code was improved by means of a new implementation in C++ and CUDA, resulting in the Iber+ (García-Feal et al., 2018). This improved model allowed for a much higher efficiency of the simulations achieving a speed-up of two-orders of magnitude while maintaining the same precision. This was possible by using GPU (graphical processing unit) computing and HPC (high performance computing) techniques. This reduction in computation times allows dealing with large spatial domains and long-term events, thus providing a high capacity to simulate flood events with high accuracy (García-Feal et al., 2018; Fernández-Nóvoa et al., 2020; Bermúdez et al., 2021; Bonasia and Ceragene, 2021; González-Cao et al., 2021). An executable Iber+ version is freely available for download from its official website (https://iberaula.es).

In the present study, Iber+ was used to study flood events in the lower Tagus valley and to analyze the effectiveness of several dam operating strategies in terms of flood mitigation. For that, the domain defined in Iber+ includes the Tagus basin from Almourol to its mouth in Mar da Palha (the large basin in the estuary of the Tagus River near its mouth) at Lisbon (Figure 1). The inlet is defined as a Critical/Subcritical condition, which allows representing the real conditions of the river using the daily mean river flow estimated in Almourol as input, whereas the outlet of the domain is defined by means of a Supercritical/Critical condition, which allows the drainage of water without altering the flow regime, providing an adequate representation of river downstream (Iber user's manual in https://www.iberaula.es/54/iber-model/downloads). Both conditions were successfully applied in previous studies where flood hydraulic simulations were also performed (Fernández-Nóvoa et al., 2020; Santillán et al., 2020; González-Cao et el., 2021; 2022). Precipitation data from Iberia01 were included in the domain being the infiltration computed by means of SCS-CN (Soil Conservation Service Curve Number) methodology (Mockus, 1964; Beven, 2012), which is especially suitable and widely applied to address rainfall-runoff computations in precipitation events (Wang, 2018; Fernández-Nóvoa et al., 2020), and according to data provided by GCN250 (Jaafar et al., 2019). The spatial patterns of different land uses were defined using CORINE land cover data, which provides a minimum mapping unit of 25 hectares with a minimum width of 100 m (CLC, 2000). The land uses for the area under study, and the associated roughness manning coefficients that were used in the model, were detailed in figure 2.

The entire domain was discretized using a mesh of unstructured triangles with an average side length varying from 25m in the flood valley to 100m in the rest of the domain, surpassing 6M elements. The computational time step is variable and was calculated following the Courant-Friedrichs-Levy (CFL) condition (Courant et al., 1967). The Courant number was set to 0.45 and the average computational time step varied between 0.05s-0.55s for the different simulations.

Regarding the model simulations, it is also important to comment that the aggradational and degradational processes were not considered in the simulations, due to the lack of available data and also considering the macroscopic scope of the research, covering the entire lower Tagus valley, which prevent to analyze in detail multiple processes. In the same way, the inline structures such as bridges were not considered in the simulations. We acknowledge that in further studies addressing local areas with more detail by using higher resolution modelling, these issues should be included and evaluated in the flood 160 analysis.

Several simulations were used here. The first (*Simulation_Control_1979*) is focused on reproducing the spatial extension and depth of the flood observed in the lower Tagus section in the 1979 event, considering the historical timing and magnitude of water released by the main dams upstream as well as the precipitation downstream. The results obtained with this simulation were also used to validate the accuracy of the model by comparing with the available information on this flood. The 165 remaining simulations (*Simulation_Dam*), deal with artificial changes imposed on the dams with the aim of mitigating the flood magnitude in the lower Tagus area.

**4.3 Digital Elevation Models (DEMs)**

Different widely used and tested freely available global DEMs (Mukherjee et al., 2013; Szabó, 2015; Becek et al., 2016; Carrera-Hernandez, 2021; Guth and Geoffroy, 2021) were evaluated to select the most suitable one to reproduce the floods 170 in the area under scope. The analyzed DEMs were: i) ESRI-DEM for Portugal mainland was made available by ESRI Portugal (ESRI copyright ©) through the ArcGIS online platform (downloadable from www.arcgisonline.com/home/search.html?t=content&q=owner:ESRI-PT). This DEM is based on data obtained by the Terra-ASTER (Advanced Spaceborn Thermal Emission and Reflection Radiometer) sensor adapted to Portugal; ii) ASTER-GDEM obtained by photogrammetric methods from the Japanese ASTER-VNIR sensor (infra-red nadir and backwards 175 sensors with 15 m GSD) and provided by NASA Earth Data and from Japan Space Systems (downloadable from the USGS https://earthexplorer.usgs.gov/); iii) SRTM-DEM obtained by the Shuttle Radar Topography Mission by SAR Interferometry (downloadable from the USGS, https://earthexplorer.usgs.gov/); iv) Copernicus DEM (COP-DEM GLO-30) provided by the European Space Agency (ESA) and AIRBUS. This DEM is based on the WorldDEM™, which is in turn based on edited and smoothed radar satellite data acquired during the TanDEM-X mission (downloadable from ESA's Copernicus Space 180 Component Data Access (CSCDA) system, https://panda.copernicus.eu/). Some original characteristics of these DEMS are specified in Table S1 of Supplementary Material. All the databases are freely available and provide horizontal resolutions about of 30m.

At this point it is important to note that, although the events to be simulated occurred several years ago, and the DEMs tested are more recent, their use is appropriate for the purposes of this study. Despite the fact that some changes may have occurred at local scale, the macroscopic picture of the terrain topography is similar. In addition, there is a lack of precise and well distributed older terrain information to perform an accurate reconstruction of the terrain elevation for the periods under interest. In fact, previous studies analyzing past events at very local scale in nearby areas, also used current elevation information due to the limited changes in terrain elevation along with the lack of precision of older data (González-Cao et al., 2021; 2022). Furthermore, the flood events analyzed span from several years between them, and a common framework is necessary to compare and evaluate the differences in flood impact under the mitigation strategies proposed, as well as in order to be applied in further studies. Hence the use of the available current DEMs.

**4.4 Validation method of the hydraulic model coupled with different DEMs.**

The flood event of 1979 was simulated with the Iber+ model coupled to each of these different DEMs to evaluate their performance to represent flooding in lower Tagus valley, taking advantage of the information provided by SNIRH database on the maximum water level reached during this event at some points located throughout the area under scope (see black circles in Figure 1). For that, the maximum water levels reached at these control points were extracted for the simulation carried out with each DEM. Thus, real maximum water levels were compared with water levels obtained in each simulation performed using the different DEMs under analysis. With this purpose, Taylor diagrams (Taylor, 2001) were used to make this comparison and, therefore, test the performance of the hydraulic model coupled with the different DEMs. These diagrams provide a concise statistical summary of the degree of correspondence between simulated and observed fields through the joint representation of the respective statistical parameters (Taylor, 2001). It should be noticed that this methodology is especially suitable and widely used to analyze the performance of models in relation to the observations (González-Cao et al., 2019; Wijayarathne and Coulibaly, 2020; Muñoz et al., 2022). The normalized standard deviation (Eq. 1), the normalized centered root mean square difference (Eq. 2) and the correlation coefficient (Eq. 3) were used.

$$\sigma_n = \frac{\sqrt{\frac{\sum_{i=1}^{N}(S_i - \bar{S})^2}{N}}}{\sigma_O} \qquad (1)$$

$$E_n = \frac{\sqrt{\frac{\sum_{i=1}^{N}[(S_i - \bar{S}) - (O_i - \bar{O})]^2}{N}}}{\sigma_O} \qquad (2)$$

$$R = \frac{\sum_{i=1}^{N}[(S_i - \bar{S})(O_i - \bar{O})]}{N\,\sigma_S \sigma_O} \qquad (3)$$

where $S$ is the simulated water level obtained with the numerical model, $O$ is the observed water level, *barred variables* refer to mean values, $N$ is the total number of observed data, subscript $i$ refers to the different points of available data, subscript $n$ refers to normalized values, and $\sigma$ is the standard deviation.

## 4.5 Alcántara dam and optimal operating strategies for flood mitigation

### 4.5.1 The role of Alcántara dam in lower Tagus valley

Alcántara dam has by far the greatest water storage capacity in the Tagus basin, which together with its location, exerts an important regulation of river flow in lower Tagus valley, as commented above. Therefore, the evaluation of possible flood mitigation in the lower Tagus by dams will be focused on Alcántara functioning. To better know the impact of the very large Alcántara dam releases in the lower Tagus valley, a comparison between dam outflow and river flow reaching Almourol was established. It was detected that, for the common period of available data (1973-2019), Alcántara provides, on average, 59.60% of water reaching Almourol. In addition, it is known that the lower Tagus starts to overflow when it exceeds approximately 1500 $m^3s^{-1}$ (Ramos and Reis, 2001; Rebelo et al., 2018). Thus, in terms of flood analysis, this means that a dam release of approximately 1000 $m^3s^{-1}$ would imply a flood risk situation. Therefore, we will consider a flood flow at Alcántara dam when the outflow exceeds 1000 $m^3s^{-1}$. Obviously, this is a general approximation since the instantaneous flow that reaches the lower Tagus valley depends on the particular conditions downstream, and therefore, the percentage of contribution from Alcántara dam can fluctuate over time. However, the present approach, considering the flow of Alcántara as 60% of the total flow that reaches the lower Tagus valley, will be used to analyze the dam functioning in terms of flood mitigation (the corresponding flow in lower Tagus valley, that is, *Alcantara_Outflow/0.6*, will be considered as the input of the respective *Simulation_Dams*).

### 4.5.2 Development of optimal dam operating strategies for flood mitigation in the lower Tagus valley

In a first approach, a general dam operating strategy was proposed to provide a controlled outflow focused on mitigating floods. To be operational, the strategy must be sustained in a clear sequence of logical principles, such as keeping an average water storage similar to the historical one. In this sense, it is important to remind that dams are multi-purpose water resource systems, and a balance should be struck between ensuring flood mitigation and other purposes, such as water supply or hydropower production (Lee et al., 2009). Hence, the need to maintain a storage similar to the real one must be equally considered. In fact, previous studies also highlight the need to maintain dam volume conditions similar to the real ones in order to develop useful dam operating strategies (Shrestha and Kawasaki, 2020). In this sense, Alcántara dam had a mean annual volume of 62 % during the available data period (1970-2019), slightly lower (59 % on average) during the rainy season (from November to March). In this context, to maintain this dam filling level around 60% seems to represent a good compromise, not undermining the normal operability of the dam. Therefore, dam level remains at 60% occupancy as long as river flow allows it. Thus, the proposal is based on two principles: i) to maintain a dam fill level that allows a certain free

dam capacity to deal with peak flows, whenever possible, hereafter referred to base filling level (BFL); ii) the maximum outflow will be limited to 1000 m$^3$s$^{-1}$ whenever allowed by dam capacity. This value corresponds to the security outflow level considered, that is an outflow below the threshold that the downstream channel is capable of safely conveying. Similar approaches have been used by previous studies developing dam release rules based on safe flow limits (e.g. Lei et al., 2018).

The dam operation under the approach rules defined above was carried out starting in October 1970 and extended to the entire available period (September 2019) forced with the real inflow. Thus, dam volume will vary according to the differences between inflow and outflow. Controlled outflow is obtained by means of the following equation:

$$Q_o = \begin{cases} 0 & if \quad V_{d-1} \leq V_{60} \\ Q_l & if \quad V_{60} < V_{d-1} \ and \ V_{d-1} + V_i \leq V_T \\ \max[\,Q_l, Q_f\,] & if \quad V_{d-1} + V_i > V_T \end{cases} \qquad (4)$$

where $Q_o$ is the controlled outflow, $Q_l$ is the security outflow level (1000 m$^3$s$^{-1}$), $Q_f$ is the dam outflow necessary to not exceed the dam capacity, that is, to maintain the dam full (if the dam is already full it would correspond to $Q_i$, the river inflow), $V_i$ is the inflow volume, $V_{d-1}$ is the dam volume of the previous day, $V_T$ is the total capacity of the dam and $V_{60} = 0.6 \times V_T$ (corresponding to BFL = 60%). In addition, a function is applied to avoid abrupt differences between the outflow of the first two proposed outflow conditions. The overall approach used here corresponds to the Operational Strategy 1 (OS1 from now on). The flowchart defining OS1 operation is presented in figure 3.

The application of OS1 is focused on avoiding flooding as long as possible, without considering the possible peaks that may arrive later. For that, a dam operating strategy more focused on mitigating the most extreme peak flows, those that can cause catastrophic floods like those recorded in 1979, was also proposed. This also allows comparing the overall implications of different approaches of dam operation. Thus, an efficient approach mostly focused on minimizing the effects of the extreme peak flows could be to allow higher outflows, above 1000 m$^3$s$^{-1}$, in the previous days, guaranteeing a sufficient dam free volume to minimize the extreme peaks, which would result in producing controlled floods. However, it is important to take into account that the outflow must always be limited by the inflow when it exceeds the safety flow, that is, the outflow can never exceed the inflow under flood conditions. This allows reducing the extreme peaks but avoiding inducing man-made floods, an approach also considered fundamental in previous studies addressing dam operation to mitigate floods (Chou and Wu, 2015). To achieve this goal, a second approach was added to OS1 to deal with extreme flow situations. Analyzing the existing data, it will only be necessary to apply this approach for the most extreme events, that is, when the expected accumulated volume for the following 7 days exceeds the 99.9$^{th}$ percentile of the historical series. Under these conditions, controlled outflow is obtained by means of the following equation:

$$Q_o = \begin{cases} 0 & if \quad V_{d-1}+V_i \le V_{60} \\ Q_l & if \quad V_{60} < V_{d-1}+V_i \le V_{90} \\ \max[Q_l, \min(Q_{o90}, Q_i)] & if \quad V_{90} < V_{d-1} + V_i \text{ and } V_i \ne V_{max} \\ \max[Q_l, \min(Q_{o100}, Q_i)] & if \quad V_{90} < V_{d-1} + V_i \text{ and } V_i = V_{max} \end{cases} \quad (5)$$

$V_{90}$ is the volume considered as the security Base Filling Level for extreme events, considered as 90% of dam capacity ($V_{90} = 0.9 \times V_T$). $V_{max}$ is referred to the day when the peak of the event is expected. $Q_{o90} = Q_i + (V_{d-1} - V_{90})x(\frac{10^6}{60x60x24})$ is the outflow which allows maintaining the volume of the dam at 90% of its capacity and $Q_{o100} = Q_i + (V_{d-1} - V_T)x(\frac{10^6}{60x60x24})$ is the outflow that allows not to exceed the dam capacity. A function is applied to avoid abrupt differences between the outflow of the first two proposed outflow conditions. The overall approach used here corresponds to the Operational Strategy 2 (OS2 from now on). The flowchart defining OS2 operation is presented in figure 4.

Thus, the present condition is focused in guaranteeing a certain free reservoir capacity to smooth the most extreme peak flows, so it will only be applied in very extreme conditions, since the necessary condition to be applied is highly restrictive. In the rest of the cases, OS1 would be applied.

We acknowledge that one important caveat of OS2 is that in order to detect possible extreme situations, it is necessary to know approximately the expected volume for the following days, however the uncertainty associated to strategies based on more precise forecasts is reduced. In fact, currently, new approaches based on the analysis and forecast of atmospheric structures that transport large amounts of moisture (such as atmospheric rivers), which are responsible for most of the extreme and large intense precipitation events, namely in the western Iberian Peninsula (Ramos et al., 2015; 2020), can provide better predictability and detection of these possible extreme situations (Ramos et al., 2020). This allows to apply the proposed strategy efficiently. However, we acknowledge that further analysis should be necessary to assess the uncertainty associated to these forecasts and their application to this case.

Finally, we consider it is important to comment that, although the strategies developed in the present study may be highly efficient for the case under scope, previous studies also developed and applied other different approaches also effective in developing dam operating strategies for flood mitigation in other locations. Thus, for example, Chou and Wu (2015) showed a methodology, applied to a dam in Taiwan, based on developing operating rules that consider 3 flood stages: prior to flood arrival, preceding flood peak and after the flood peak. Each stage has its owns rules, being the determination of the stages based on real-time measurements. While the rules intend to be based on real-time measurements, and depend as little as possible on forecast, preliminary detection (forecast) of the flood events is also necessary to initialize the process. Lei et al. (2018) showed an example of interconnected rules to operate several flood reservoirs in China, through different optimization approaches. On the one hand, a single-objective optimization was proposed to mitigate the flood in a specific

location (for example, the most vulnerable city). In this sense, all the dams operate focused on minimizing flood in this particular location. On the other hand, a multi-objective optimization was proposed to mitigate flooding in several vulnerable areas. In return, the flood mitigation in the most vulnerable location is less effective than with the single-objective optimization. Hasebe and Nagayama (2002) demonstrated that the application of neural networks for decision support could provide efficient dam operation to flood control in Japan. Other approaches were based on reproducing different hypothetical situations, including distinct ranges of river flow, as well as different conditions for dams, specifically focused on the initial water level and the degree of opening of the gates. This approach is used by some authors that show it allows providing information about the best dam operating under different flood conditions (Hardesty et al., 2018; Ridolfi et al., 2019). In this sense, other authors reproduced historical floods with hydrodynamic models, associated to critical dam releases, also carrying out studies on the resulting flooded areas with and without protection structures, such as levees, under different dam outflows, providing useful information to help in the development of mitigation strategies (Patel et al., 2017). We acknowledge that although the dam operating strategies proposed in the present work serve as a basis for developing future studies focused on optimizing dam strategies for the area under scope, some of these other approaches could also be tested and applied in future studies.

### 4.5.3 Evaluation of the performance of dam operating strategies developed

The efficiency of both proposed strategies (OS1 and OS2) was evaluated analyzing firstly the floods that would occur considering the natural regime (dam inflow) as well as the floods under the real dam operation, and then evaluating the mitigation achieved with the proposed strategies throughout the entire study period. Then, special attention will be focused on those more extreme events to analyze the efficiency of the dam operating strategies in these critical situations. In particular, five extreme cases were selected according to literature and to the available data series (http://www.cedex.es/): February 1972, March 1978, February 1979, December 1989 and November 1997. All these five cases are characterized by peak flows higher than 5000 $m^3s^{-1}$ at Alcántara location. In addition, most of them are also included in the top 10 rank of the most important extreme events in terms of daily and accumulated precipitation considering the entire Tagus basin (Ramos et al., 2014; 2017). A hydraulic analysis was also made for these most extreme events, in order to analyze the effective flood reduction achieved in lower Tagus valley by means of the two strategies proposed (OS1 and OS2). The analysis will be especially focused on evaluating the reduction achieved in flood extent, water depth and water velocity. In this sense it is important to highlight that water depth is a critical and dangerous factor in these events, since the damages caused by floods are closely linked to the water depths reached (Tsakiris, 2014; Huizinga et al., 2017). In addition, the velocity reached by water is also another factor that can increase flood damage (Cox et al., 2011).

## 5 Results and Discussion

### 5.1 Validation of different DEMs

Unlike other countries such as Spain, Portugal has only freely available high resolution Digital Elevation Models (DEMs) for the coastal area and a few additional areas (https://www.ign.es/web/ign/portal; https://dados.gov.pt/pt/). Therefore, it is no surprise that for the lower Tagus (depicted in Figure 1) a high-resolution DEM is not freely available, as these can generally only be acquired from national organizations that produce topographical and bathymetric cartography at scales compatible with local studies. This could represent a limitation to analyze in detail the flood development over concrete locations, namely within towns or villages where high resolution is necessary to adequately address the flood progress through their streets. However, global DEMs can provide an adequate representability to analyze the large-scale evolution and the magnitude of flood events (Yan et al., 2013; Courty et al., 2019). Available global DEM products have spatial resolutions varying from meters to kilometers. Horizontal and vertical accuracies can also vary greatly depending on the type of topography, and they can reach tens of meters (Thompson et al., 2001; Zhang et al., 2014).

In this context, and considering the wide range of available DEMs it was felt necessary to evaluate the suitability of different freely available DEMs to adequately represent floods in the lower Tagus valley. To achieve this goal, one of the most important flood events occurred in that area on February 1979, was simulated and analyzed for different DEMs in order to test which one is most appropriate for the area under scope. As was mentioned above, four DEMs were tested, namely ESRI, ASTER, SRTM and Copernicus DEMs (Karlsson and Arnberg, 2011; Wang et al., 2012; Garrote, 2022).

In general terms, the results obtained with Copernicus, SRTM and ASTER DEMs clearly indicate better performance for simulating floods in lower Tagus valley with respect to ESRI DEM, which provides worse results in all the statistics analyzed by means of the Taylor diagram (Figure 5). Especially highlight the results obtained with Copernicus DEM, which are clearly the closest to the reference data, indicating that Copernicus DEM presents the best accuracy, i.e. the best capability to address floods in the area under scope. In particular, it presents a high correlation with the measured data, above 0.99, with a normalized standard deviation close to 1 and the lowest normalized centered root mean square difference (< 0.1). The SRTM DEM also presents a correlation above 0.99, although the normalized standard deviation (1.11) and the normalized centered root mean square difference (0.17) are worse than those obtained with Copernicus DEM. ASTER DEM presents statistics slightly worse than SRTM DEM. In addition, the original elevation data from these DEMs were also compared with the official altimetric values by calculating several statistical indicators to evaluate the associated error and deviation, including the Mean Absolute Error (MAE), Standard Deviation (SD), Root Mean Squared Error (RMSE) and the Mean Error (ME) (see Table S2 in the Supplementary Material). Copernicus DEM is also corroborated as the most accurate, presenting the best values in all the analyzed statistics, followed again by the SRTM DEM (see the detailed analysis provided in the Supplementary Material). Additionally, the spatial distribution of the standard deviation of the absolute error for the Copernicus DEM was further investigated (Figure S1 in the Supplementary Material). The results confirm that

overall, Copernicus DEM displays low error values throughout the study area (see detailed analysis in the Supplementary Material). Recent studies comparing the accuracy of different DEMs along the European continent (Guth and Geoffroy, 2021) and in other parts of the world (Garrote, 2022), also confirm the higher precision of Copernicus DEM in comparison with other global products.

This confirms that Copernicus DEM, coupled with the Iber+ model, are capable of adequate reproduction, at large-scale, of the flood events in the lower Tagus. In fact, the statistical parameters analyzed by means of the Taylor diagrams corroborate not only the better performance compared to the other DEMs analyzed, but also the accurate representation of the reference flood data. Therefore, Copernicus DEM was selected for the remaining of the analysis.

## 5.2 Flood event of 1979

This event was simulated (*Simulation_Control_1979*) in order to analyze the spatial-temporal evolution of the flood (Figure 6). In general terms, previous studies detected that the high precipitation rates that occurred during the event and the previous days caused an increase in the Tagus River flow until it reached a relative maximum surpassing 5000 $m^3s^{-1}$ at Almourol on February 5, an amount that has been associated to the beginning of the significant impacts by flood (Zêzere et al., 2014; Rebelo et al., 2018). As it can be observed in Figure 6a, the simulation indicates that a large part of the valley is flooded under these conditions, although the water depth values are relatively low near the villages. On the following days, river flow experienced a certain decrease, and consequently, the simulation shows a decreasing water depth, with water slightly receding from some locations (Figure 6b). However, the situation degrades again afterwards and starts to be critical from February 9 on, reaching an outstanding maximum around February 11, surpassing 13000 $m^3s^{-1}$ at Almourol. Under these conditions most of the valley is flooded but the most important fact detected by the simulation is that the water depth reveals a significant increase (Figure 6c). The high water levels reached near the villages in the simulation are a clear indication that they were affected by the flood, as confirmed by previous studies (Zêzere et al., 2014; Rebelo et al., 2018). These publications corroborate the flood impact on several important villages located in the valley, including V. N. da Barquinha, Golegã, Chamusca, Santarém, and Vila Franca de Xira (Figure 1). As commented above, the resolution of the available DEMs does not allow resolving the entire topographic characteristics of the villages, however results obtained by the simulation support the flood of these locations. In addition, previous analysis showed that these high water levels implicated the isolation of some of these village populations and, in several occasions people had to be evacuated by boats and helicopters since roads or railroads were waterlogged (Zêzere et al., 2014; Rebelo et al., 2018). Although these structures usually present a certain elevation in respect to the surrounding terrain, the water depths reached during this event implicated that they were equally flooded (Rebelo et al., 2018). Overall, the reports of flooded locations and specific sites are supported by the model results. In particular, the simulation indicates an increase in the water level of the Tagus River in Santarem above 4 meters respect to the non-flood situation, a similar increase detected with the measured data (Rebelo et al., 2018), stressing the unusual level of this flood event. Although the river flow decreased in the following days, the flood situation

still lasted. Our simulation indicates that, on February 16 (Figure 6d), although water levels suffer an important decrease, and water recedes from some locations of the valley, a large faction of the valley remained flooded.

There are some documents that collect detailed information related to the maximum water depths reached during this flood, constituting a valuable additional source of data useful to validate the accuracy of the model. First of all, the Portuguese National Civil Engineering Laboratory (LNEC) and Water Institute (INAG) reconstructed the maximum flooding area
during this event (Figure 7, white line). As can be observed in Figure 7, simulated flood extension is practically coincident with the registered flood extension. Only in the areas inside the towns, flood extension provided by the simulation is a little smaller due to the limitations of the DEMs commented above for which the advance of water in towns is limited.

More specifically, Rebelo et al. (2018) stated that the railroad line near to Golegã was flooded and important means of transport were interrupted. As it can be observed in Figure 8 (panel 1), the simulation detected that this infrastructure was
395 flooded over a large section, corroborating the blocking of the railroad. Another location with valuable information of the flood is the town of Benfica do Ribatejo. In particular, some photographs warrant the flood of the football field of this location, situation also reproduced in the model (Figure 8, panel 3). In the surroundings of Santarem, specifically in the statue dedicated to Santa Iria (Figure 8, panel 2), there is information about the depth reached by water during this event. The existing documents indicate that water almost reached the feet of the statue (Loureiro, 2007), which indicate a water depth
about of 3m. In the model simulation water also reaches this area (Figure 8, panel 2), with water depths surpassing the 2.5m, which is in good accordance with real situation. Additionally, Figure 8 (panel 4) represents Palhota town. In this location there are several water marks in different houses indicating an approximate water depth of 1.8 meters. In the simulation this town is also affected by floods, as can be detected in Figure 8 (panel 4), with a water depth ranging between 1 and 2 meters in the surrounding of this area, in line with the *in situ* measurements. This larger range of values reflects the particular
location of this water mark, inside the town where the global DEMs presents higher uncertainty. Therefore, it can be concluded that the simulations carried out in this work, using Iber+ model in combination with Copernicus DEM, allow a good reproduction of the 1979 flood extent and water depth in the lower Tagus valley. However, we acknowledge that there are some caveats in our modelling, namely the relatively low resolution of our DEM, specially over constructed areas that have been flooded and where a more detailed analysis requires a higher resolution DEM such as the DEM produced and
made available by the Direção Geral do Território (DGT), which corresponds to a strip of 600 m at sea and 400 m on land, of the coastal areas of mainland Portugal with a resolution of 2 m, obtained from a survey with LiDAR technology. An equivalent DEM, but for all continental territory, was already announced to be in construction in 2021 by DGT, although it is not yet available at the time of this study. Thus, although the DEM used appears to provide an adequate macroscopy view of the flood, and allows the general analysis of flood mitigation under the different dam strategies presented below, the absolute
values obtained in some locations should be taken with caution for the reasons commented above.

### 5.3 Mitigation of Tagus floods

### 5.3.1 General analysis of the effectiveness of the dam operating strategies proposed in relation to flood reduction

Firstly, a general analysis of historical functioning of Alcántara dam in relation to floods, together with the functioning applying OS1, was carried out. For that, the number of days with outflows with high probability of causing flooding downstream (> 1000 m$^3$s$^{-1}$) was evaluated considering: i) the inflow, which represents an approximation to the natural regime of the Tagus River (from now on considered as the natural regime); ii) the real dam outflow, which corresponds to what really happened; iii) a controlled outflow provided by OS1.

The results of the three approaches, natural regime, real dam outflow and controlled outflow by the dam operating proposed, are summarized in Table 1. Considering a natural regime, Tagus River flow at Alcántara higher than the considered security level was observed 445 days (average of 9.1 days per year). The actual dam regulation over these years promoted an important reduction in these risk flows, with 42 % less of cases, with a total number of 260 (5.3 days per year). The reduction in risk flows achieved with the OS1 strategy is much higher, with a total number of days under flood conditions of 74 (1.5 days per year), i.e. a decrease of 83 % in the number of cases compared to the corresponding number for natural flow. In addition, OS1 also provides an effective reduction of the most critical river flows, reducing by 53 % the likelihood of cases in which the daily flows are exceed 3000 m$^3$s$^{-1}$ and by 50 % those days with flows greater than 5000 m$^3$s$^{-1}$. Thus, under the OS1, the mean dam volume along the simulation is practically the same as in the reality, but the number of floods decreases considerably.

Having demonstrated the soundness of OS1 in decreasing the number of days under flood flow conditions, the question that remains is whether that approach is sufficient to prevent flooding in the most extreme cases. The inflow and the controlled outflow (following OS1) are represented in Figure 9 for the five extreme cases under study. Although, the number of days under flood conditions is significantly reduced for the extreme events analyzed, (7(0), 14(6), 27(10), 28(12), 27(0)), extreme flood conditions are not completely prevented. In the events with higher river flows that persists for several days, the most extreme peak flows are not smoothed, as observed in the cases of the events in 1978, 1979 and 1989. Therefore, an improvement is needed to also address these most extreme cases. In particular, considering as a benchmark the flood that occurred in 1979, it can be observed that with the application of the OS1, the simulated outflow coincides with the natural regime (inflow) for the maximum peak flow, as well as for the following days (Figure 9, third panel), indicating that the dam was full. When the dam operating strategy focused on extreme peak flows is applied (OS2), dam functioning can improve the efficiency to mitigate extreme peaks, as shown in Table 2, that provides the main characteristics of the controlled dam outflow for the extreme cases. It can be noticed that the main peak flow is reduced by about 30% in the 1979 flood event. An important reduction also occurred in the other extreme cases equally not smoothed with OS1: the peak flow of the 1978 flood event is reduced by more than 25 %, and the peak of the 1989 extreme event is reduced by about 40 %. In fact, when the series along the entire period under study is analyzed, it can be observed that the OS2 provides a greater reduction in the

number of high peak flow cases ($>3000$ m$^3$s$^{-1}$ and $>5000$ m$^3$s$^{-1}$, see Table 1). In return, the total number of days under flood conditions ($> 1000$ m$^3$s$^{-1}$) is slightly increased when OS2 is applied in relation to OS1, due to the fact that under OS2 flows above the flood threshold are allowed prior to extreme peaks in order to minimize them, as commented above. In any case, when applying OS2 the total number of days under flood conditions is also significantly reduced compared to natural regime and real dam outflow (see Tables 1 and 2).

Having demonstrated the efficiency of the proposed dam operating strategies considering the actual conditions of river flow for the period under scope, the series of river flow was perturbed through the addition of noise, in order to analyze the applicability of the proposed strategies in other scenarios of river flow. For that, perturbed random series were generated allowing a deviation of $\pm$ 25 % from the original values of dam inflow, that is, each real daily value of river flow has been allowed a random variation of $\pm$ 25 %. Following this procedure, as many perturbed series as the original number of data were generated ($> 17000$). Then, the average number of floods generated by the river flow of the perturbed series (hypothetical natural regime), as well as the respective floods resulting from applying the dam operating strategies proposed, were evaluated (Table 3). The efficiency of both proposed strategies was clearly maintained in terms of reducing the total number of floods. In fact, even considering the worst possible situation taking into account the associated deviation in each case, floods are significantly reduced. Moreover, the efficiency of OS2 to mitigate the most extreme floods was also maintained. The obtained results corroborate the robustness and the applicability of dam operating strategies proposed under different scenarios of river flow.

## 5.3.2 Hydraulic analysis of the effectiveness of the dam operating strategies in relation to flood mitigation

It is important to take into account that although the peaks are reduced applying the OS2, the total multi-day outflow volume throughout the events is similar (within each event) for the dam operations considered. Therefore, it is crucial to assess what both approaches imply in terms of effective flooding reduction in lower Tagus valley. This is especially relevant taking into account the flattened shape of the valley, which implies that a large area can be flooded, even with relatively lower flows as it was detected in Figure 6. This issue can be addressed again taking advantage of the Iber+ hydraulic model. Therefore, three simulations (*Simulation_Dam*) were performed for each extreme event, that is, considering outflow at Alcántara corresponding to: i) natural regime, ii) OS1; iii) OS2. To keep the analysis within a manageable size we restrict the full assessment to the flood event of 1979 considered as benchmark, where Figure 10a shows the natural river flow and the controlled outflow of the dam obtained by applying the configuration focused on the extreme events (OS2). In addition to the reduction of maximum amplitude, the peak flow is delayed by one day, which can further decrease flood damage downstream. To evaluate the real reduction on flooding in lower Tagus valley for the event of 1979 under the different strategies presented, the simulated maximum flood caused by the Alcántara outflow resulting from natural regime and operating strategy OS2 is shown in Figures 10b and 10c, respectively. The most important fact is that the entire area presents an important reduction in water depth under the most effective dam operating strategy. This reduction is shown in more

detail in Figure 11 where the differences between both cases are highlighted, with a decrease that can surpass one meter in some locations. In addition, although the reduction in flood extension is small compared to the total extension of the valley, it can be detected as water is also retracted at some extent in the surroundings of the villages (see zoomed areas in Figure 11). Therefore, the application of the OS2 could suppose an important flood alleviation for the area under scope.

This flood mitigation analysis was also applied to all the other extreme events, in order to provide information focused on
effective mitigation measures and to understand their impact on flood reduction. For that, the floods caused by the different configurations, that is, the natural flow regime and the operating strategies OS1 and OS2, were analyzed and compared for the most critical events by means of the respective hydraulic simulations (Table 4). This allows to extract key information for the area under scope. As commented above, the events occurred in 1972 and 1997 can be completely avoided at Alcántara location, therefore the analysis will be focused on the rest of the extreme events. According to hydraulic
simulations, in these most extreme events is not possible to prevent most of the valley from being flooded even with dam regulation presented in OS2. This is mainly due to the flattened shape of the valley, which favors flooding even with lower discharges, as commented above. Specifically, a reduction of around 5-10% in the total extension of the flood is achieved on average in the most extreme cases (from 564 km$^2$ to 535 km$^2$ in the 1979 flood event). Once most of the valley is flooded, the main increase occurs in terms of water depth. In this sense, the mitigation of floods in the lower Tagus valley allowed by
an efficient regulation of dams, is especially effective in terms of water depth reduction (Table 4). On average, OS2 allows reducing water depths in flooded areas by more than 0.5 m, which supposes a reduction of around 25 % in water depth. In the particular case of the 1979 flood, the average maximum water depth is reduced from 2.56 m to 1.94 m. This contributes to a significant reduction in flood damages and the associated costs, which are proportional to the water depths (Huizinga et al., 2017), as commented above. Another important aspect that should be taken into account to assess flood damages is the
maximum velocity reached by water (Cox et al., 2011). In this sense, the presented strategy also contributed to a reduction in this flood hazard metric, reducing the maximum velocity reached by the water in the flooded areas by around 25-30 % (Table 4). In particular, the average maximum velocity of the water in flooded areas is reduced from 0.51 ms$^{-1}$ to 0.38 ms$^{-1}$ in the 1979 flood.

The results obtained corroborate the important mitigation of flood impacts that can be achieved in the lower Tagus basin
taking advantage of the existing dam capacity.

## 6 Summary and Conclusions

This work aimed to present dam operating strategies that allow taking advantage of existing infrastructures in the Tagus River to effectively mitigate floods, that have occurred in its lower valley in recent decades, and may occur again in the future. For this, dam operating strategies were developed and, in combination with the Iber+ hydraulic model, the
effectiveness of the proposal in relation to flood mitigation was analyzed. To perform this analysis, the other important

objective achieved was the validation of the hydraulic model for the lower Tagus valley by evaluating its ability to reproduce the 1979 flood.

Thus, firstly, Iber+ model was validated for the area under scope. In this process, several DEMs were used to also determine the best one to macroscopically reproduce the floods in the lower Tagus valley. Copernicus DEM shows the best accuracy.
In fact, Iber+ model coupled with Copernicus DEM was able to provide an adequate macroscopic reproduction of the most important flood of the last 150 years in the Tagus valley, the 1979 flood, which demonstrates its capability to evaluate floods in the area under study and allows the hydrodynamic analysis of this event. This also provides information of robust and useful tools that could serve as a basis to future studies addressing other different aspects related to flooding in lower Tagus valley.

Once the Iber+ model was validated, the analysis was focused on developing dam operating strategies to help in flood mitigation. Specifically, the analysis was focused on Alcántara dam, the most important on the Tagus River. In general terms, results indicate that the first proposed strategy (OS1) allows diminishing the number of days under flood conditions by more than 80 % with respect to the natural regime, and an important reduction is also obtained in relation to the historical dam operation. In addition, the mitigation of the most extreme flood events was also achieved. Hydraulic simulations
confirm that the proposed operating strategy focused on the mitigation of extreme events (OS2) is especially effective in reducing water depth and water velocity in the flooded areas (~ 25-30 %), the most critical factors in terms of flood damage. In addition, a smaller reduction in flood extension is also achieved (~ 5-10 %). Therefore, hydraulic simulations corroborate the significant flood mitigation in the lower Tagus valley that can be achieved with more appropriate use of dam strategies, as proposed in this work. This demonstrates the effectiveness of the strategies proposed to address the future implications of
climate change in relation to the expected more frequent and intense flood events in the future. Thus, the mitigation strategies OS1 and OS2 represent an example of a set of non-expensive strategies that will allow the mitigation of floods taking advantage of the existing infrastructures, and that can serve as example of application to other basins.

In summary, this study can be viewed as a first step to improve the knowledge on extreme floods in the lower Tagus valley and to provide strategies to mitigate these events taking advantage of the existing infrastructures, thus addressing one of the
most important challenges that the scientific community will have to face in the coming decades as a consequence of climate change. Future improvements should be focused, on the one hand, on the development of similar strategies applied to other important dams throughout the Tagus basin, both in the Spanish and Portuguese sections, providing a cascading interconnection between the different dams and the operation strategies developed, which will improve and make more effective the flood mitigation provided by these infrastructures. On the other hand, future improvements should also consider
the integration of the dam operating strategies for real-time early warning systems. These strategies, in combination with the hydraulic models and good weather forecasts, will allow evaluating in advance the likelihood of flood scenarios and apply the right measures that minimize the floods (Chang et al., 2010; Chou and Wu, 2015; Fakhruddin et al., 2015; Fraga et al., 2020). This will also allow to improve and make more precise the dam strategies applied. Furthermore, although the model

used showed sufficient capacity to simulate floods in the lower Tagus valley on a large scale, we acknowledge that
improvements should be done in order to analyze in detail flood impact in local areas such as villages or specific
infrastructures. In particular, the development of high resolution DEMs would be essential as it allows for more detailed
definition of the terrain topography. In consequence, it would also be advisable a more detailed information on other features
such as river and flood plain roughness, along with a specific evaluation of the most precise model parameters for higher
resolutions (flood cell size, Courant number…). The inclusion of detailed local structures that can affect flood development
would also be welcome. These improvements will contribute to improve the precision of the simulations, needed to address
the impact at local scale.

*Code and data availability:* Freely available data and software were used for this work. Regarding the hydrodynamic model
used, it should be noted that although the open-source code is only accessible to collaborators, an executable Iber+ version is
freely available for download from its official website (https://iberaula.es).

*Author contribution:* DFN: Conceptualization, Methodology, Formal analysis, Investigation, Writing – Original Draft.
AMR: Methodology, Investigation, Writing – Review & Editing. JGC: Methodology, Investigation, Writing – Review &
Editing. OGF: Methodology, Investigation, Writing – Review & Editing. CC: Methodology, Investigation, Writing – Review
& Editing. MGG: Conceptualization, Methodology, Writing – Review & Editing, Supervision. RMT: Conceptualization,
Methodology, Writing – Review & Editing, Supervision.

*Competing interests:* The authors declare that they have no conflict interest.

**Acknowledgements**

The authors thank the "Sistema Nacional de Informação de Recursos Hídricos" (SNIRH), the Centro de Estudios y
Experimentación de Obras Públicas (CEDEX), the developers of Iberia01 database and the respective developers of the
DEMs used, for the information provided for this work. The authors thank Google for the courtesy of provide some of the
aerial maps used in this work.
This research has been partially supported by Xunta de Galicia, Consellería de Cultura, Educación e Universidade, under
Project ED431C 2021/44 "Programa de Consolidación e Estructuración de Unidades de Investigación Competitivas". This

research has also been partially supported by the European Regional Development Fund under the INTERREG-POCTEP project RISC_PLUS (Code: 0031_RISC_PLUS_6_E)

DFN was supported by Xunta de Galicia through a post-doctoral grant (ED481B-2021-108). AMR was supported by the Helmholtz "Changing Earth" program. CC was supported by EEA-Financial Mechanism 2014-2021 and the Portuguese Environment Agency through Pre-defined Project-2 National Roadmap for Adaptation XXI (PDP-2). OGF was funded by Spanish "Ministerio de Universidades" and European Union – NextGenerationEU through the "Margarita Salas" post-doctoral grant. RMT was supported by the Portuguese Science Foundation (FCT) through the project AMOTHEC (DRI/India/0098/2020).

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

**Table and Figure Captions**

**Table 1.** Number of days under different critical flows at Alcántara location considering the real inflow (natural regime), the real outflow, and the dam outflow under the operation strategies OS1 and OS2. Percentages are referred to the differences with respect to the worst scenario (natural regime), which is assigned a percentage of 100 %.

**Table 2.** Hydrologic characteristics of most extreme flood events under different dam configurations: NR (natural regime - no dam), OS1 and OS2 (operation strategy 1 and 2 in equation (4) and (5), respectively). Flood days refer to the number of days exceeding the flood threshold. Peak flow refers to the real maximum daily inflow in the case of the natural regime, and to the maximum daily outflow from the dam under the different operation strategies. Percentages are referred to the differences with respect to the worst scenario (natural regime), which is assigned a percentage of 100 %.

**Table 3.** Average number of days (and the corresponding standard deviation) exceeding different critical flows at Alcántara location, calculated taking into account all the perturbed river flow series (created by varying randomly in ±25% the original dam inflow). Natural regimes, and dam outflows obtained by applying the operation strategies OS1 and OS2, were evaluated.

**Table 4.** Hydraulic characteristics of most extreme flood events in the lower Tagus valley considering the outflows provided by the different dam configurations considered (*Simulation_Dam*): NR (natural regime ), OS1 (operation strategy 1), and OS2 (operation strategy 2). The percentage values represent the reduction obtained with respect to the situation under natural regime.

**Figure 1.** Area of study. Panel a) indicates the location of the study area (dashed black rectangle) including the lower Tagus valley. Green diamond indicates the location of Almourol station and green circle indicates the location of Alcántara dam. In panel b), the black circles represent the control points where there is data on the water levels reached in the 1979 flood event. The numbered grey triangles represent areas with relevant flood information (particular flooded areas, water depths…) for the 1979 event: 1 - railroad of Golegã, 2 - Santa Iria statue at Santarem, 3 - football field at Benfica do Ribatejo, and 4 - Palhota town. The numbered white circles indicate the location of the main villages affected by the flood: 1 - V.N. da Barquinha, 2 - Golegã, 3 - Chamusca, 4 - Santarém and 5 - Vila Franca da Xira. In panel c), the main locations of interest are represented. Bathymetry and topography basemaps were provided by ESRI©. Sources: Esri, GEBCO, NOAA, National Geographic, Garmin, HERE, Geonames.org, and other contributors; Esri, Garmin, GEBCO, NOAA NGDC, and other contributors.

**Figure 2.** Land uses, from CORINE land cover, for the study area, and associated manning coefficients.

**Figure 3.** Flowchart of the dam operation strategy presented in equation (4): OS1. $Q_o$ is the controlled outflow, $Q_l$ is the security outflow level (1000 $m^3s^{-1}$), $Q_f$ is the dam outflow necessary to not exceed the dam capacity, that is, to maintain the dam full (if the dam is already full it would correspond to $Q_i$, the river inflow), $V_i$ is the inflow volume, $V_{d-1}$ is the dam

volume of the previous day, $V_T$ is the total capacity of the dam and $V_{60} = 0.6 \times V_T$ (corresponding to BFL = 60%). In addition, a function is applied to avoid abrupt differences between the outflow of the first two proposed outflow conditions.

**Figure 4.** Flowchart of the dam operation strategy presented in equation (5): OS2. $V_{90}$ is the volume considered as the security Base Filling Level for extreme events, considered as 90% of dam capacity ($V_{90} = 0.9 \times V_T$). $V_{max}$ is referred to the day when the peak of the event is expected. $Q_{o90} = Q_i + (V_{d-1} - V_{90}) x (\frac{10^6}{60 x 60 x 24})$ is the outflow which allows maintaining the volume of the dam at 90% of its capacity and $Q_{o100} = Q_i + (V_{d-1} - V_T) x (\frac{10^6}{60 x 60 x 24})$ is the outflow that allows not to exceed the dam capacity. A function is applied to avoid abrupt differences between the outflow of the first two proposed outflow conditions.

**Figure 5.** Taylor diagram of the water elevation obtained with Iber+ using the field data as reference. E, A, S and C indicate the Iber+ data obtained using the ESRI, ASTER, SRTM and Copernicus Digital Elevation Models, respectively.

**Figure 6.** Reproduction of water depth (meters) for the flood event occurred in February, 1979 in lower Tagus valley, using Iber+ hydraulic model. a), b), c) and d) represents the flood situation on 5, 8, 11 and 16 February under the Simulation_Control_1979. Map data © Google Satellite.

**Figure 7.** Maximum flood extension for event of February, 1979, in lower Tagus, obtained from the hydraulic simulation (Simulation_Control_1979). The white line represents the real extension of the flood reconstructed by the National Civil Engineering Laboratory (LNEC) and the Water Institute (INAG) from Portugal (www.snirh.pt). Map data © Google Satellite.

**Figure 8.** Detailed flooded area obtained with hydraulic simulation (Simulation_Control_1979) for: 1) railroad of Golegã, 2) Santa Iria statue at Santarem, 3) football field at Benfica do Ribatejo, and 4) Palhota town. The red arrow indicates the level reached by the water. The photographs and measurements in panels 2) and 4) were taken by the authors. Basemap from Google Terrain Hybrid. Aerial maps in panels 1), 2), 3) and 4) from: Map data © Google Satellite.

**Figure 9.** Natural regime (blue line) and simulated outflow resulting for the operation strategy OS1 (red line) for Alcántara dam in the most extreme cases. The date (month) refers to the occurrence of the highest peak flow. The initial date considered for each event is, from top to bottom: January 19, 1972; February 11, 1978; January 18, 1979; November 15, 1989; October 31, 1997.

**Figure 10.** (a) Natural regime (dam inflow) at Alcántara (blue line) and simulated Alcántara dam outflow under the operation strategy OS2 (red line), considering the flooding of 1979. Lower panels show the maximum water depth (meters) obtained with Iber+ for the outflows corresponding to (b) natural regime, and (c) dam operation strategy OS2. Map data © Google Satellite.

**Figure 11.** Difference in maximum water depth (meters) caused by the Alcántara outflows corresponding to natural regime (NR) and operation strategy OS2 (OS2 – NR), applied to the 1979 flood event. Red colors represent locations reached by water under the most extreme case (NR) and not flooded when OS2 is applied. Left zoomed area represents the surroundings of Castanheira do Ribatejo town, whereas right zoomed area represents the zone delimited by the towns of Mato de Miranda, Azinhaga and Pombalinho in the surroundings of Golegã location. Map data © Google Satellite.

| Parameter | Natural Regime (dam inflow) | Real Dam Outflow | Operation Strategy OS1 | Operation Strategy OS2 |
|---|---|---|---|---|
| *Days > 1000 m³s⁻¹* | 445 (100 %) | 260 (58 %) | 74 (17 %) | 83 (19 %) |
| *Days > 3000 m³s⁻¹* | 32 (100 %) | 24 (75 %) | 15 (47 %) | 13 (41 %) |
| *Days > 5000 m³s⁻¹* | 8 (100 %) | 5 (63 %) | 4 (50 %) | 1 (13 %) |

**Table 1.** Number of days under different critical flows at Alcántara location considering the real inflow (natural regime), the real outflow, and the dam outflow under the operation strategies OS1 and OS2. Percentages are referred to the differences with respect to the worst scenario (natural regime), which is assigned a percentage of 100 %.

| Event | Flood Days | | | Peak Flow (m³s⁻¹) | | |
|-------|------|------|------|------|------|------|
|       | *NR* | *OS1* | *OS2* | *NR* | *OS1* | *OS2* |
| *1972* | 7 | 0 | 0 | 5170 (100 %) | 1000 (19 %) | 1000 (19 %) |
| *1978* | 14 | 6 | 6 | 5513 (100 %) | 5513 (100 %) | 4003 (73 %) |
| *1979* | 27 | 10 | 12 | 7965 (100 %) | 7965 (100 %) | 5545 (70 %) |
| *1989* | 28 | 12 | 14 | 5593 (100 %) | 5593 (100 %) | 3332 (60 %) |
| *1997* | 27 | 0 | 0 | 5301 (100 %) | 1000 (19 %) | 1000 (19 %) |


**Table 2.** Hydrologic characteristics of most extreme flood events under different dam configurations: NR (natural regime - no dam), OS1 and OS2 (operation strategy 1 and 2 in equation (4) and (5), respectively). Flood days refer to the number of days exceeding the flood threshold. Peak flow refers to the real maximum daily inflow in the case of the natural regime, and to the maximum daily outflow from the dam under the different operation strategies. Percentages are referred to the
differences with respect to the worst scenario (natural regime), which is assigned a percentage of 100 %.


| Parameter | Natural Regime | Operation Strategy OS1 | Operation Strategy OS2 |
|---|---|---|---|
| *Days > 1000 m³s⁻¹* | 445.85 ± 7.51 | 75.58 ± 3.32 | 84.36 ± 3.46 |
| *Days > 3000 m³s⁻¹* | 34.23 ± 2.92 | 15.69 ± 1.87 | 13.94 ± 2.05 |
| *Days > 5000 m³s⁻¹* | 6.72 ± 1.57 | 3. 42 ± 1.01 | 1.52 ± 0.95 |

**Table 3.** Average number of days (and the corresponding standard deviation) exceeding different critical flows at Alcántara location, calculated taking into account all the perturbed river flow series (created by varying randomly in ±25 % the original dam inflow). Natural regimes, and dam outflows obtained by applying the operation strategies OS1 and OS2, were
evaluated.

| Event | Maximum Flooded Area (km²) | | | Mean Water Depth (m) | | | Mean Water Velocity (m/s) | | |
|---|---|---|---|---|---|---|---|---|---|
| | *NR* | *OS1* | *OS2* | *NR* | *OS1* | *OS2* | *NR* | *OS1* | *OS2* |
| *1978* | *540.24* | 531.32 **-1.7 %** | 504.17 **-6.7 %** | *2.03* | 1.90 **-6.4 %** | 1.53 **-24.6 %** | *0.40* | 0.37 **-7.5 %** | 0.30 **-25.0 %** |
| *1979* | *564.22* | 561.37 **-0.5 %** | 535.45 **-5.1 %** | *2.56* | 2.53 **-1.2 %** | 1.94 **-24.2 %** | *0.51* | 0.50 **-2.0 %** | 0.38 **-25.5 %** |
| *1989* | *534.39* | 532.43 **-0.4 %** | 486.71 **-8.9 %** | *1.93* | 1.91 **-1.0 %** | 1.40 **-27.5 %** | *0.38* | 0.37 **-2.6 %** | 0.27 **-28.9 %** |

**Table 4.** Hydraulic characteristics of most extreme flood events in the lower Tagus valley considering the outflows provided by the different dam configurations considered (*Simulation_Dam*): NR (natural regime), OS1 (operation strategy 1), and OS2 (operation strategy 2). The percentage values represent the reduction obtained with respect to the situation under natural regime.

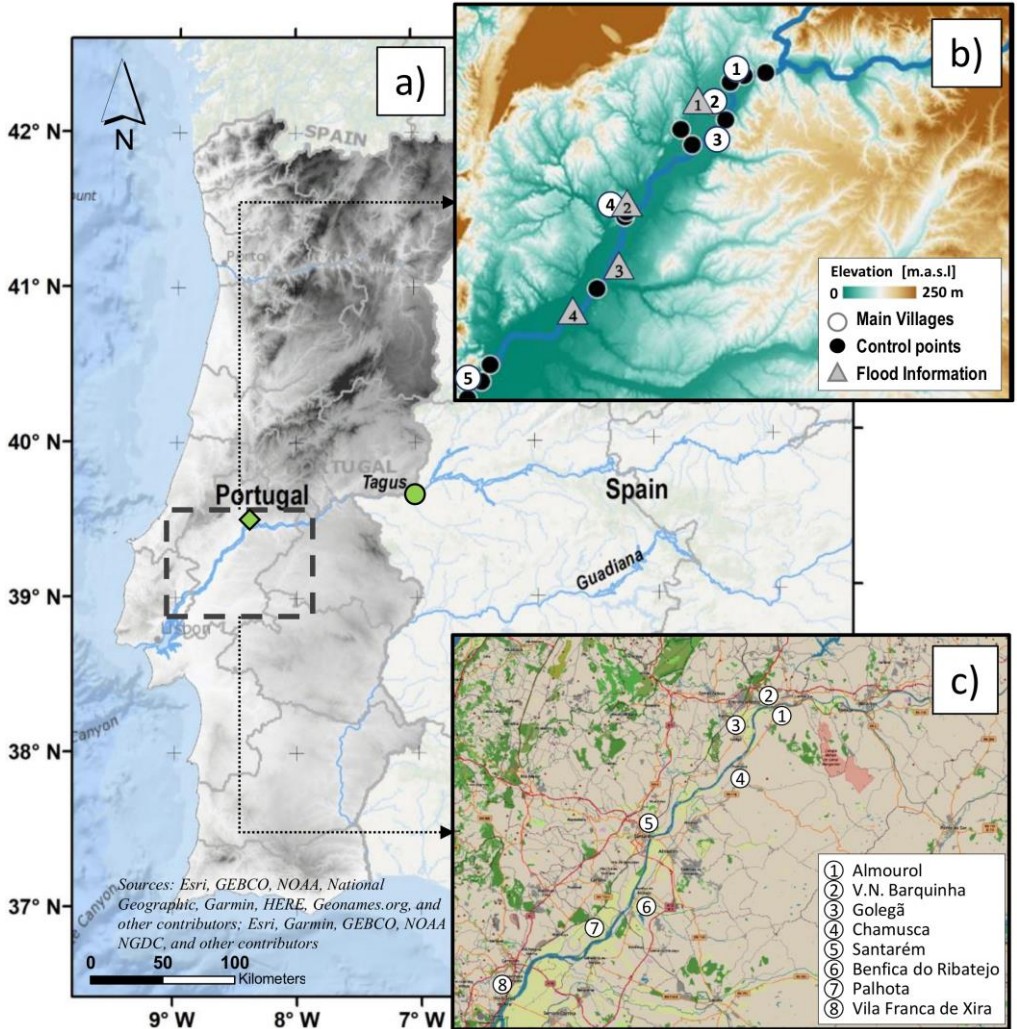

**Figure 1.** Area of study. Panel a) indicates the location of the study area (dashed black rectangle) including the lower Tagus valley. Green diamond indicates the location of Almourol station and green circle indicates the location of Alcántara dam. In panel b), the black circles represent the control points where there is data on the water levels reached in the 1979 flood event. The numbered grey triangles represent areas with relevant flood information (particular flooded areas, water depths…) for the 1979 event: 1 - railroad of Golegã, 2 - Santa Iria statue at Santarem, 3 - football field at Benfica do Ribatejo, and 4 -
Palhota town. The numbered white circles indicate the location of the main villages affected by the flood: 1 - V.N. da Barquinha, 2 - Golegã, 3 - Chamusca, 4 - Santarém and 5 - Vila Franca da Xira. In panel c), the main locations of interest are represented. Bathymetry and topography basemaps were provided by ESRI©. Sources: Esri, GEBCO, NOAA, National Geographic, Garmin, HERE, Geonames.org, and other contributors; Esri, Garmin, GEBCO, NOAA NGDC, and other contributors.


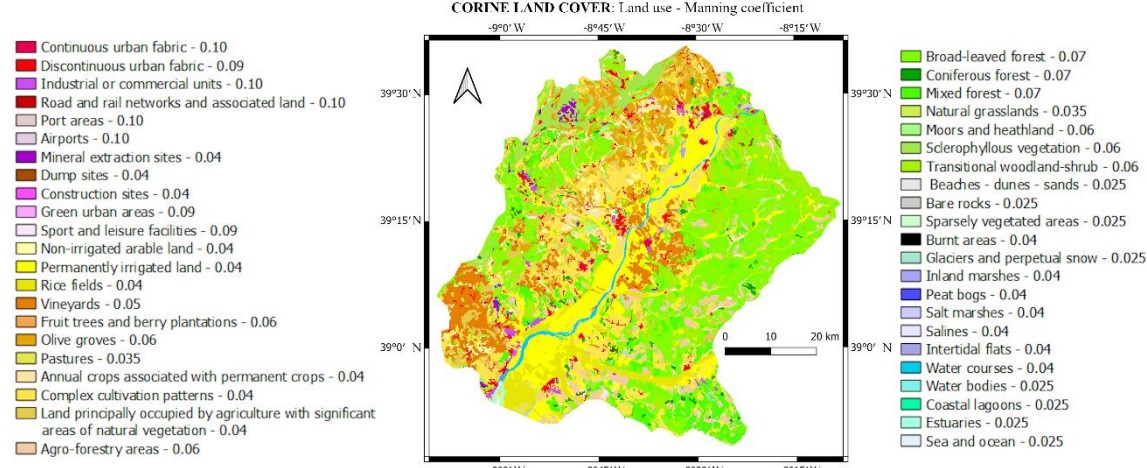

**CORINE LAND COVER**: Land use - Manning coefficient

Continuous urban fabric - 0.10
Discontinuous urban fabric - 0.09
Industrial or commercial units - 0.10
Road and rail networks and associated land - 0.10
Port areas - 0.10
Airports - 0.10
Mineral extraction sites - 0.04
Dump sites - 0.04
Construction sites - 0.04
Green urban areas - 0.09
Sport and leisure facilities - 0.09
Non-irrigated arable land - 0.04
Permanently irrigated land - 0.04
Rice fields - 0.04
Vineyards - 0.05
Fruit trees and berry plantations - 0.06
Olive groves - 0.06
Pastures - 0.035
Annual crops associated with permanent crops - 0.04
Complex cultivation patterns - 0.04
Land principally occupied by agriculture with significant areas of natural vegetation - 0.04
Agro-forestry areas - 0.06

Broad-leaved forest - 0.07
Coniferous forest - 0.07
Mixed forest - 0.07
Natural grasslands - 0.035
Moors and heathland - 0.06
Sclerophyllous vegetation - 0.06
Transitional woodland-shrub - 0.06
Beaches - dunes - sands - 0.025
Bare rocks - 0.025
Sparsely vegetated areas - 0.025
Burnt areas - 0.04
Glaciers and perpetual snow - 0.025
Inland marshes - 0.04
Peat bogs - 0.04
Salt marshes - 0.04
Salines - 0.04
Intertidal flats - 0.04
Water courses - 0.04
Water bodies - 0.025
Coastal lagoons - 0.025
Estuaries - 0.025
Sea and ocean - 0.025

**Figure 2.** Land uses, from CORINE land cover, for the study area, and associated manning coefficients.

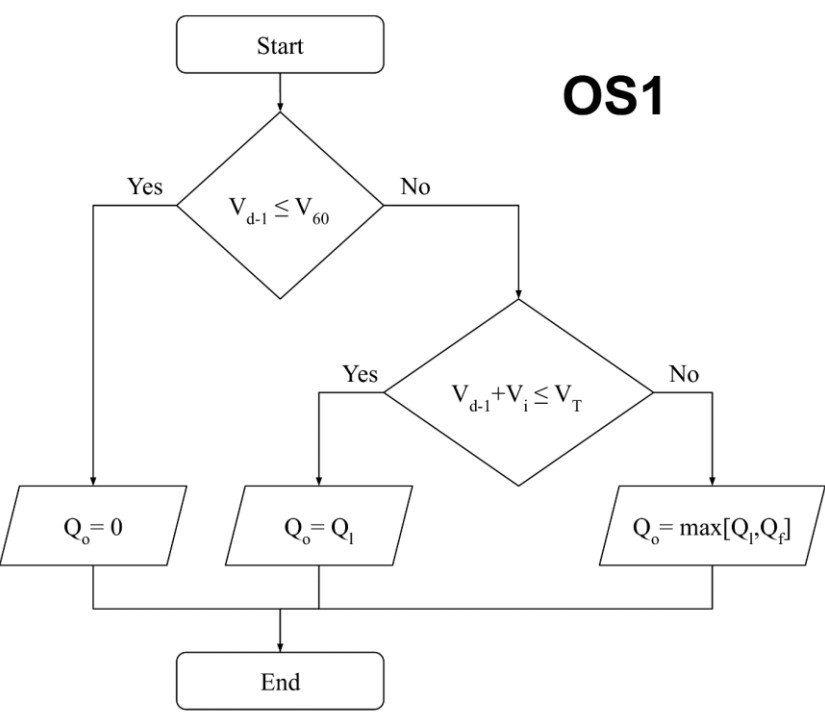

**Figure 3.** Flowchart of the dam operation strategy presented in equation (4): OS1. $Q_o$ is the controlled outflow, $Q_l$ is the security outflow level (1000 m$^3$s$^{-1}$), $Q_f$ is the dam outflow necessary to not exceed the dam capacity, that is, to maintain the dam full (if the dam is already full it would correspond to $Q_i$, the river inflow), $V_i$ is the inflow volume, $V_{d-1}$ is the dam volume of the previous day, $V_T$ is the total capacity of the dam and $V_{60} = 0.6 \times V_T$ (corresponding to BFL = 60%). In addition, a function is applied to avoid abrupt differences between the outflow of the first two proposed outflow conditions.

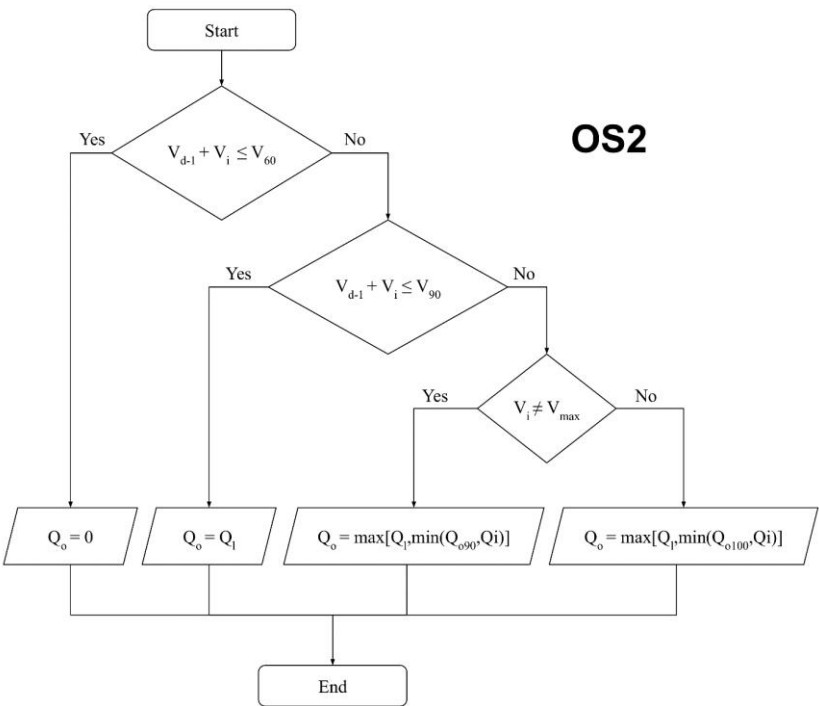

**Figure 4.** Flowchart of the dam operation strategy presented in equation (5): OS2. $V_{90}$ is the volume considered as the security Base Filling Level for extreme events, considered as 90% of dam capacity ($V_{90} = 0.9 \times V_T$). $V_{max}$ is referred to the day when the peak of the event is expected. $Q_{o90} = Q_i + (V_{d-1} - V_{90})x(\frac{10^6}{60x60x24})$ is the outflow which allows maintaining the volume of the dam at 90% of its capacity and $Q_{o100} = Q_i + (V_{d-1} - V_T)x(\frac{10^6}{60x60x24})$ is the outflow that allows not to exceed the dam capacity. A function is applied to avoid abrupt differences between the outflow of the first two proposed outflow conditions.

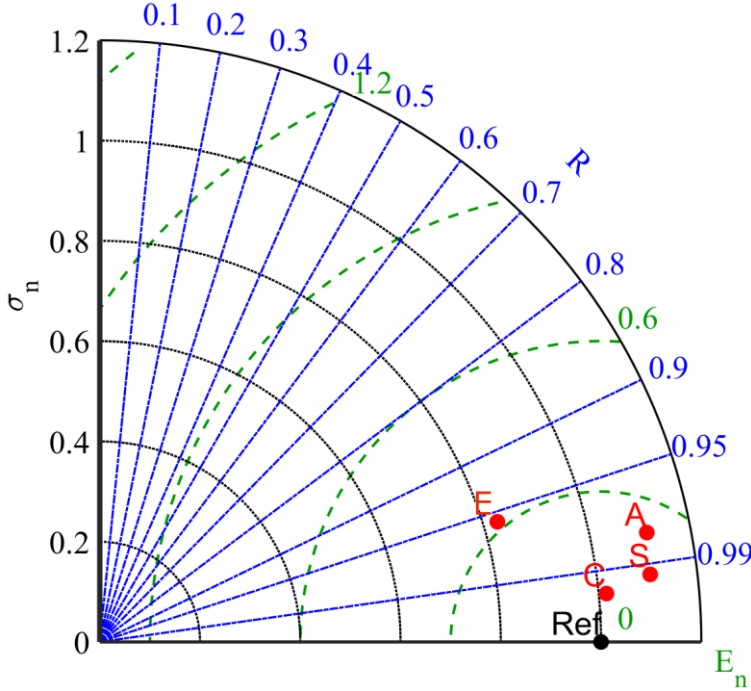


**Figure 5.** Taylor diagram of the water elevation obtained with Iber+ using the field data as reference. E, A, S and C indicate the Iber+ data obtained using the ESRI, ASTER, SRTM and Copernicus Digital Elevation Models, respectively.


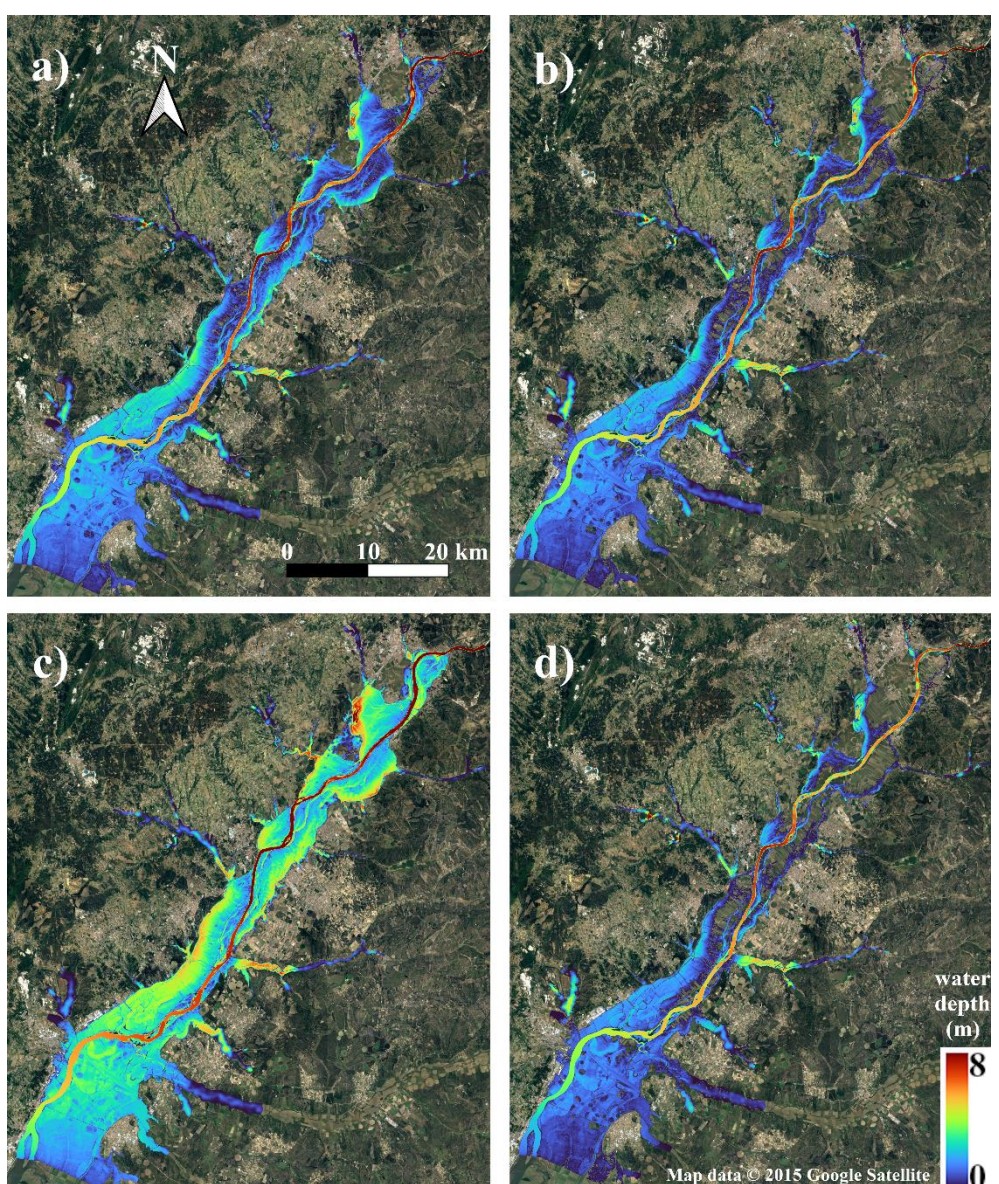


**Figure 6.** Reproduction of water depth (meters) for the flood event occurred in February, 1979 in lower Tagus valley, using Iber+ hydraulic model. a), b), c) and d) represents the flood situation on 5, 8, 11 and 16 February under the Simulation_Control_1979. Map data © Google Satellite.


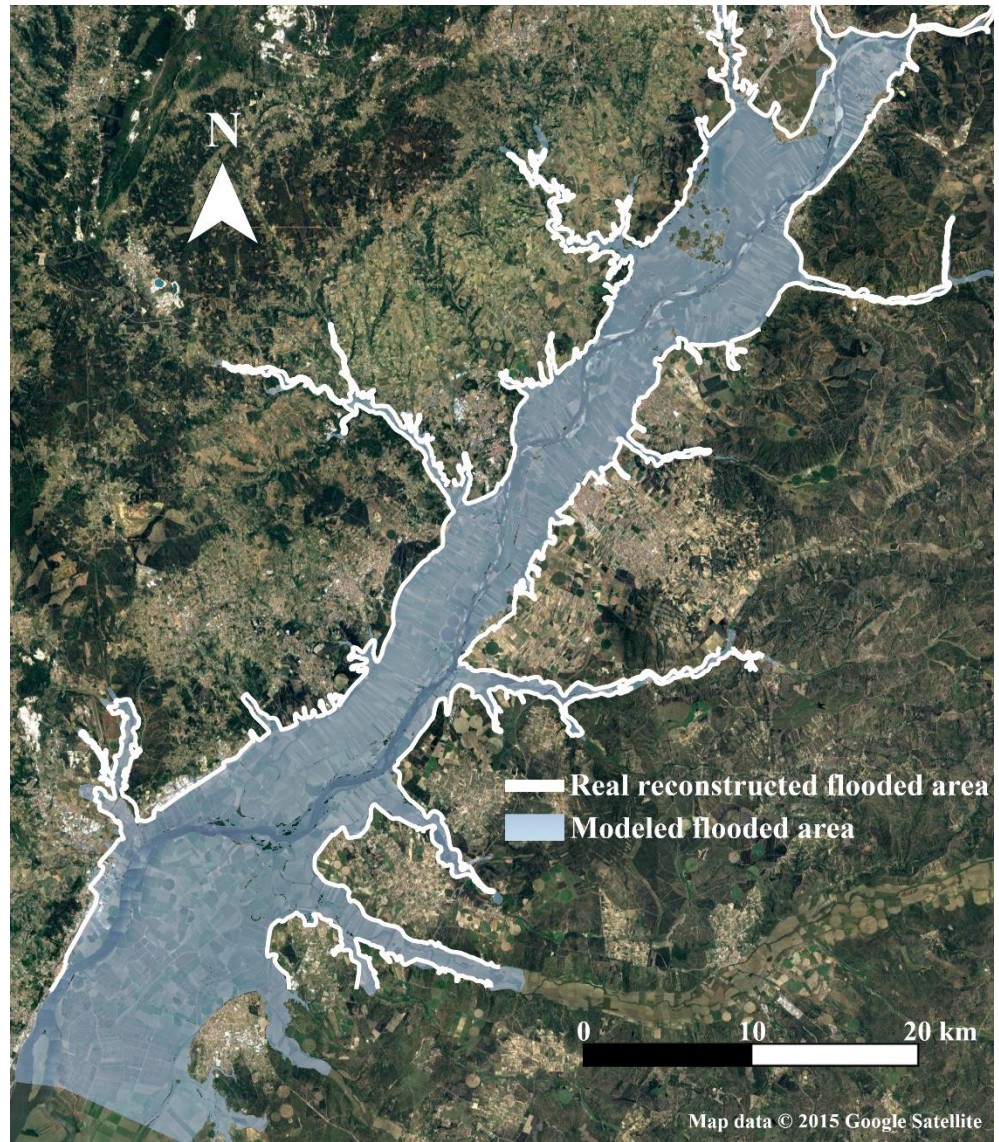

**Figure 7.** Maximum flood extension for event of February, 1979, in lower Tagus, obtained from the hydraulic simulation (Simulation_Control_1979). The white line represents the real extension of the flood reconstructed by the National Civil Engineering Laboratory (LNEC) and the Water Institute (INAG) from Portugal (www.snirh.pt). Map data © Google Satellite.

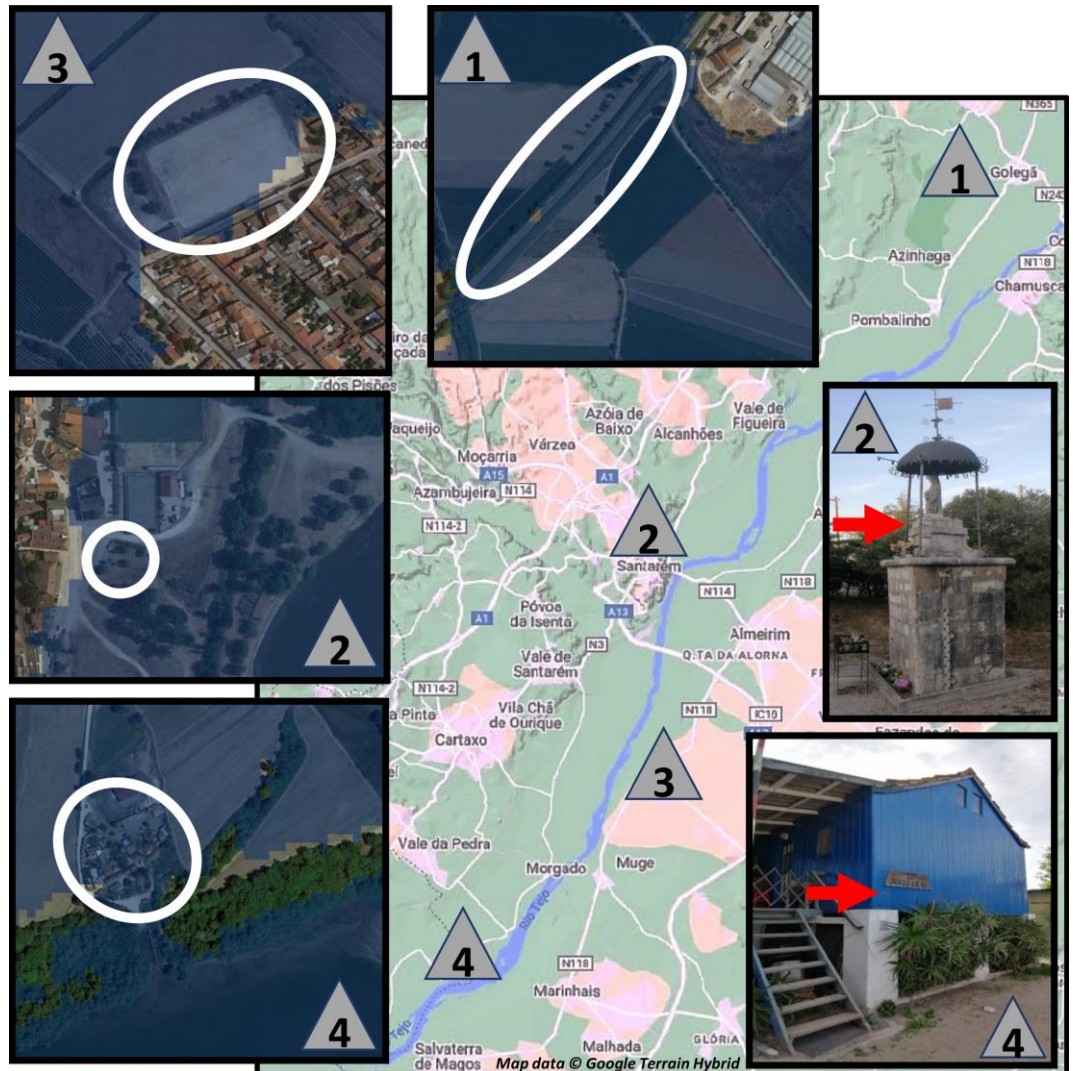

**Figure 8.** Detailed flooded area obtained with hydraulic simulation (Simulation_Control_1979) for: 1) railroad of Golegã, 2) Santa Iria statue at Santarem, 3) football field at Benfica do Ribatejo, and 4) Palhota town. The red arrow indicates the level reached by the water. The photographs and measurements in panels 2) and 4) were taken by the authors. Basemap from Google Terrain Hybrid. Aerial maps in panels 1), 2), 3) and 4) from: Map data © Google Satellite.

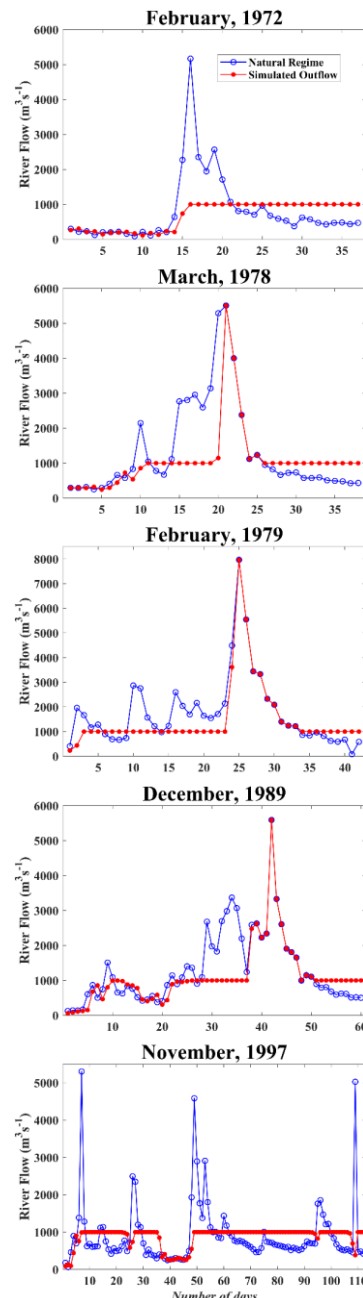

**Figure 9.** Natural regime (blue line) and simulated outflow resulting for the operation strategy OS1 (red line) for Alcántara dam in the most extreme cases. The date (month) refers to the occurrence of the highest peak flow. The initial date considered for each event is, from top to bottom: January 19, 1972; February 11, 1978; January 18, 1979; November 15, 1989; October 31, 1997.

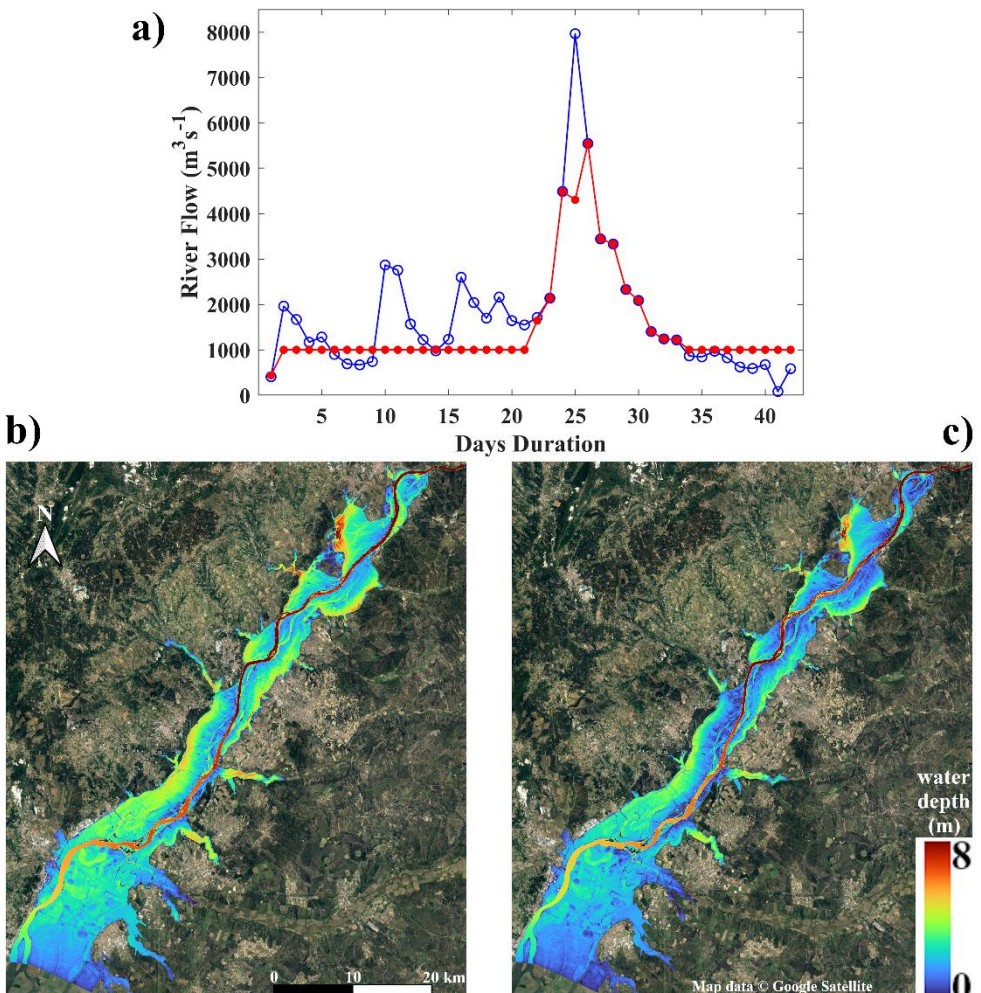

**Figure 10.** (a) Natural regime (dam inflow) at Alcántara (blue line) and simulated Alcántara dam outflow under the operation strategy OS2 (red line), considering the flooding of 1979. Lower panels show the maximum water depth (meters) obtained with Iber+ for the outflows corresponding to (b) natural regime, and (c) dam operation strategy OS2. Map data © Google Satellite.

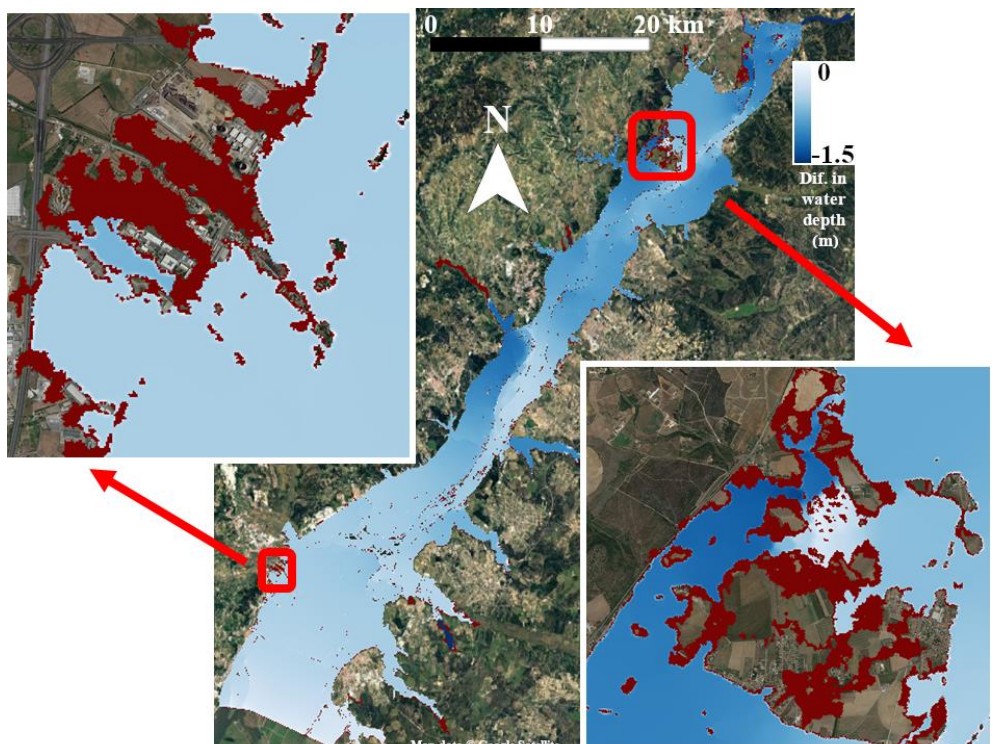

**Figure 11.** Difference in maximum water depth (meters) caused by the Alcántara outflows corresponding to natural regime (NR) and operation strategy OS2 (OS2 – NR), applied to the 1979 flood event. Red colors represent locations reached by water under the most extreme case (NR) and not flooded when OS2 is applied. Left zoomed area represents the surroundings of Castanheira do Ribatejo town, whereas right zoomed area represents the zone delimited by the towns of Mato de Miranda, Azinhaga and Pombalinho in the surroundings of Golegã location. Map data © Google Satellite.
