# Peer review of "How to mitigate flood events similar to the 1979 catastrophic floods in lower Tagus"

_Natural Hazards and Earth System Sciences, 2022_

## Referee Comment (RC2)

**nhess-2022-243**

**Title: How to mitigate flood events similar to the 1979 catastrophic floods in lower Tagus**

**Reviewer Comments:**

1. The novelty part of the MS is missing in the MS. Please mention.
2. The authors have mentioned the 2D hydrodynamic model (Iber+ numerical model). Is this model is open source? Please specify.
3. Please mention the comparative analysis of simulated model with Iber+ numerical model to check the accuracy of model.
4. The statistical analysis is missing in the MS.
5. Please do the sensitivity analysis of the model.
6. Kindly, separate the discussion section and mention the limitation and recommendation of the study.

---

## Author Comment (AC1)

First of all, the authors want to thank the referee for the work and time devoted to review the manuscript. We know that all comments will serve to improve the quality and understanding of the work and we hope we have properly answered all the suggestions.

**Reviewer #1:**

**Summary**

*The study demonstrates two operation strategies for flood mitigation in the lower Tagus valley. The Tagus is the largest river on the Iberian Peninsula and the study focuses on flood mitigation by means of regulating the largest dam of the river, the Alcántara dam. The Iber+ model is employed together with a pre-selected digital elevation model (Copernicus) to conduct the simulation of river flow and operating strategies in hindsight at the example of five flood events between 1972 and 1997 with a focus on the major flooding in 1979.*

**Evaluation and Recommendation**

*The manuscript is an interesting case study to the highly important field of flood mitigation. The topic of this manuscript is suitable for the journal.*

*The manuscript is overall well-written and extensively referenced. Data and software that were used are freely available and data sources are referenced. Appealing maps are provided as figures to illustrate the site of investigation.*

*While the content might be valuable to readers that are interested in flood mitigation operation strategies, the presentation of the material requires some restructuring and elaboration. One of the major points in this respect is that the results section includes many statements that should rather be presented in the introduction and in the methods section. In the current form, the actual results are presented between many of these additional statements and references, and key insights are therefore hard to elicit. Content-wise, the line of thought can be followed but the reader gets the impression that a step beyond is missing: As addition, it would strengthen the manuscript if there were a comparison with other operating strategies that are mentioned (see, e.g. l. 273-275) but not further pursued. Alternatively, an uncertainty analysis using noise that is added to the historic data (like perturbed precipitation data) could demonstrate the applicability of the proposed operating strategies beyond the deterministic hindsight scenario, specifically w.r.t. to the suggested operation strategy 2.*

*The manuscript bears the potential for being a valuable contribution to the field once the points laid out here are addressed. My recommendation therefore is major revision. In my comments, I suggest changes about restructuring the article. These shall outline one way to do it and do not have to be followed strictly. Yet, an iterated presentation of the material is required.*

The authors would like to thank the reviewer for the valuable comments.

Following the reviewer's suggestions, the article was restructured and some of the statements previously included in the Results and Discussion section are now included in either the "Introduction" or in the "Data, Models and Methods" sections, hopefully providing better structuring of the manuscript. Additionally, a more in depth analysis and comparison with other operating strategies are also carried out in the new version of the manuscript.

In addition, the hindsight scenario with historic river flow data was perturbed in order to test more in depth the performance of the dam operation strategies proposed. In this sense, random perturbed series were generated allowing a deviation of ± 25 % from the original values, that is, each real daily value of river flow has been allowed a random variation of ± 25 %. In this sense, as many perturbed random series as total days of the original series (> 17000) were generated to add more robustness to the results. Finally, the average number of floods generated by the river flow of the perturbed series (perturbed natural regime) as well as the associated standard deviation, were presented in a new table. Likewise, the mean number of floods (and the respective standard deviation) generated by the river flow of the perturbed series but applying operating strategies 1 (OS1) and 2 (OS2) was also evaluated and presented in the new table. In all cases, the proposed strategies provide an important reduction in the number of floods, and additionally, the efficiency of OS2 to mitigate the most extreme floods was also confirmed. This corroborates the robustness and the applicability of dam operating strategies proposed under different scenarios of river flow. This information was added in the new version of the manuscript.

| Parameter | Natural Regime | Operation Strategy OS1 | Operation Strategy OS2 |
|---|---|---|---|
| *Days > 1000 $m^3s^{-1}$* | 453.97 ± 87.63 | 77.64 ± 3.34 | 89.05 ± 3.57 |
| *Days > 3000 $m^3s^{-1}$* | 37.98 ± 3.16 | 16.13 ± 2.05 | 14.42 ± 2.28 |
| *Days > 5000 $m^3s^{-1}$* | 6.98 ± 1.61 | 3. 65 ± 0.88 | 1.66 ± 0.98 |

**Table.** The original series of real inflow at Alcántara location was perturbed by applying a random deviation of ± 25% to the daily river flow values. Thus, several random perturbed series equal to the total number of days from the original series (> 17000 days) were created. Then, the mean number of days (and the corresponding standard deviation) exceeding different critical outflows at Alcántara location, were calculated for the perturbed river flow series, considering a natural regime and also, applying the operation strategies OS1 and OS2.

Finally, the authors made an effort to make relevant information more accessible and, in particular, the proposed operating strategies were presented in flowcharts for ease of understanding in the new version of the manuscript.

[Figure]

**Figure.** Flowchart of the dam operation strategy 1: OS1. $Q_o$ is the controlled outflow, $Q_l$ is the security outflow level (1000 m$^3$s$^{-1}$), $Q_i$ is the river inflow, $V_i$ is the inflow volume, $V_{d-1}$ is the dam volume of the previous day, $V_T$ is the total capacity of the dam and $V_{60} = 0.6 \times V_T$ (corresponding to BFL = 60%).

[Figure]

**Figure.** Flowchart of the dam operation strategy 2: OS2. $V_{80}$ is the volume considered as the security Base Filling Level for extreme events, considered as 80% of dam capacity ($V_{80} = 0.8 \times V_T$). $V_{max}$ is referred to the day when the peak of the event is expected. $Q_{o80} = Q_i + (V_{d-1} - V_{80}) x (\frac{10^6}{60x60x24})$ is the outflow which allows maintaining the volume of the dam at 80% of its capacity and $Q_{o100} = Q_i + (V_{d-1} - V_T) x (\frac{10^6}{60x60x24})$ is the outflow that allows not to exceed the dam capacity.

*Specific comments*

- *Abstract: "…several dams…" is stated, but only the major Alcántara dam is regulated under historic data, right? Please make sure to not raise wrong expectations in the abstract.*

**The reviewer is correct. The statement was not entirely clear in the original abstract. We have changed this sentence to clarify that the dam operating strategies were only applied to the Alcántara dam, the most important for the Tagus river.**

- *l. 26: "…with the proposed strategies". Briefly name what is the core idea behind the strategies already in the abstract.*

**This was included in the new version of the manuscript.**

"… In this sense, dam operating strategies were developed and analyzed for the most important dam along the Tagus river basin in order to propose effective procedures to take advantage of these infrastructures to minimize the effect of floods. Overall, the numerical results indicate a good agreement with water marks and some descriptions of the 1979 flood event, which demonstrates the model capability to evaluate floods in the area under study. Regarding flood mitigation, obtained results indicate that the frequency of floods can be reduced with the proposed strategies, which were focused on providing optimal dam operating rules to mitigate flooding in lower Tagus valley"

- *Provide a clear "Motivation" or "Objective" section. The intention of the paper becomes clear over when reading the article, but it is preferable to have the goals clearly stated at some point – this is also something the final conclusions can relate to. One important motivation that is worth mentioning might be that a thought-through operation strategy is a low-cost approach for flood mitigation compared to additional structures that have to be build.*

**Done. A new section 2. Motivation, was developed.**

**"2 Motivation**

The main motivations driving this study are, on the one hand, to improve the knowledge and understanding of flood development in lower Tagus valley, an area especially vulnerable to these events. In this sense, one of the main motivations for carrying out this analysis was the scarcity of studies addressing this issue, especially from a hydrodynamic point of view. For that, different freely available products were tested in order to provide the most accurate tools that can serve as a basis for future studies focused on addressing different aspects related to flooding in lower Tagus valley. On the other hand, the study also intends to provide different strategies to mitigate floods in lower Tagus valley but taking advantage of existing infrastructures, in particular, the dams. To the best of our knowledge, there are no previous studies that

have developed this type of strategies for the area under scope, so the strategies presented in this work could represent an important advance in this field. This proposal will allow to provide an affordable new approach to flood mitigation compared to the implementation of additional structural measures that have to be built. For that, dam operating strategies will be proposed and tested in the most important Tagus dam. The benefits provided by the dam strategies proposed in relation to flood mitigation, will be also evaluated. This will also serve as a basis for developing future studies focused on optimizing dam strategies or even interconnecting the strategies of different dams of the Tagus basin to improve the flood mitigation."

- *Section 2: Please explain why the Alcántara dam is the only one considered here and whether there are other operating strategies that include other structures as well.*

**The Alcántara dam is by far the one with the largest capacity of the dams located along the Tagus basin, which, together with its location, on the border between Spain and Portugal, implies that the Tagus river flow in the Portuguese sector is, to a large extent, controlled by this dam. Therefore, taking into account that one of the main objectives of this work is to propose dam strategies to help in the mitigation of floods in lower Tagus sector, we opted to develop and apply dam operating strategies only to this dam. Thus, here we want to show an example of how an adequate regulation of this dam alone could prevent or, at least, strongly mitigate floods in lower Tagus valley. This could provide a basis for further works that can take advantage of the information and results presented here, namely to apply different strategies along the entire Tagus basin, or even interconnect the strategies of different dams, to make improved and more efficient strategies to flood mitigation in Tagus river. This was clarified in the new version of the manuscript.**

- *Section 3: rephrase as "Data, Models and Methods"*

**Done.**

- *Section 3: merge 3.1 – 3.4 in subsection "Data"*

**Done.**

- *Section 3: great public resources!*

**Thanks. We always try, as far as possible, to conduct the studies with public resources in order to facilitate the transfer of the knowledge acquired.**

- *L 92: please add which distance is resembled by 0.1° in km*

**Done.**

"0.1º ($\approx$ 10 km)".

- *L .120ff: please explain in more detail why the model was operated the chosen way: why was the inlet chosen to be critical/subcritical? Why was the outlet chosen to be supercritical/critical? Are these conditions static or do they change over the time series? Why was the SCS-CN methodology by Mockus (1964) used for getting the infiltration from precipitation – are there no more recent and potentially improved alternatives?*

**The inlet of the model was defined by means of the critical/subcritical condition because it allows reproducing the real conditions of the river flow using as input the values of the time series of Tagus river flow registered at the gauge station located in Almourol (inlet area of the numerical domain). Other types of inlet conditions depend on other parameter that can suppose an additional source of uncertainty. The outlet was defined as supercritical/critical condition since it allows reproducing with reasonable accuracy the river flow situation, taking into account that no control stations are located at this point and, therefore, no data of river conditions were recorded. Both boundary conditions (inlet and outlet) allow, in our view, a good balance between accuracy and simplicity, since other conditions can only be defined using more data that, in this case, cannot be accessed. These conditions remain static during the simulations. This kind of inlet and outlet conditions were successfully applied in other works where flood hydraulic simulations were carried out, obtaining accurate results (Fernández-Nóvoa et al., 2020; Santillán et al., 2020; González-Cao et al., 2021; 2022). This rational was further clarified in the new version of the manuscript.**

**Regarding the SCS-CN methodology, the reference of Mockus (1964) refers to the initial development of this methodology, which is currently widely used to estimate the runoff in extreme flood events (Beven, 2012; Wang, 2018; Fernández-Nóvoa et al., 2020). In the new version of the manuscript the bibliography was updated and more information about this methodology was added.**

Beven, K.: Rainfall-Runoff Modelling: The Primer, 2nd Edn., Wiley-Blackwell, Chichester, UK, 2012.

Fernández-Nóvoa, D., García-Feal, O., González-Cao, J., de Gonzalo, C., Rodríguez-Suárez, J. A., Ruiz del Portal, C., and Gómez Gesteira, M.: MIDAS: A New Integrated Flood Early Warning System for the Miño River, Water, 12, 2319, https://doi.org/10.3390/w12092319, 2020.

González-Cao, J., Fernández-Nóvoa, D., García-Feal, O., Figueira, J. R., Vaquero, J. M., Trigo, R. M., and Gómez-Gesteira, M.: The Rivillas flood of 5–6 November 1997 (Badajoz, Spain) revisited: An approach based on Iber+ modelling, J. Hydrol., 610, 127883, https://doi.org/10.1016/j.jhydrol.2022.127883, 2022.

González-Cao, J., Fernández-Nóvoa, D., García-Feal, O., Figueira, J. R., Vaquero, J. M., Trigo, R. M., and Gómez-Gesteira, M.: Numerical reconstruction of historical extreme floods: The Guadiana event of 1876, J. Hydrol., 599, 126292, https://doi.org/10.1016/j.jhydrol.2021.126292, 2021.

Mockus, V., 1964. National engineering handbook. US Soil Conservation Service. Washington, DC, USA, 4.

Santillán, D., Cueto-Felqueroso, L., Sordo-Ward, A., Garrote, L.: Influence of Erodible Beds on Shallow Water Hydrodynamics during Flood Events, Water, 12(12), 3340, https://doi.org/10.3390/w12123340, 2020.

Wang, D.: A new probability density function for spatial distribution of soil water storage capacity leads to the SCS curve number method, Hydrol. Earth Syst. Sci., 22, 6567-6578, https://doi.org/10.5194/hess-22-6567-2018, 2018.

- *L 124: which sizes: side length or circumference?*

**The side length. This was clarified in the new version of the manuscript.**

- *L 126: "…tries to reproduce…" Does it only try to reproduce? Of course it is a simulation, but please*

**The sentence was rewritten as follows:**

"Several simulations were used here. The first (Simulation_Control_1979) is focused on reproducing the spatial extension and depth of the flood observed in the lower Tagus section in the 1979 event, considering the historical timing and magnitude of water released by the main dams upstream as well as the precipitation downstream."

- *L 147: please describe why a Taylor diagram was used and how it works, i.e. that it is a tool for visualizing multi-objective optimization*

**A Taylor diagram is a very suitable tool that provides a concise statistical summary of how a pattern matches with other (Taylor et al., 2001). It allows representing multiple statistics in a compact single diagram. In particular, Taylor diagrams provide a way of plotting together three well known model validation statistics to carry out this comparison, in this case the correlation coefficient, the normalized root mean square difference and the normalized standard deviation. Thus, the correlation coefficient provides information about the similarity pattern of the target and reference series. The normalized root mean square difference allows quantifying the differences between the target and the reference series, complementing the statistical information about the correspondence between the**

different patterns analyzed. Finally, the normalized standard deviation allows completing the characterization of how a target series corresponds to the reference series. These statistical parameters are widely used in statistical analysis and they are linked to provide a very comprehensive evaluation of the results, allowing to evaluate the degree of correspondence between simulated and observed fields (Taylor et al., 2001). Thus, this diagram has been widely applied to analyze the performance of models in relation with the reality that pretends simulate (e.g., González-Cao et al., 2019; Wijayarathne and Coulibaly, 2020; Muñoz et al., 2022). This more complete description of Taylor diagrams was included in the new version of the manuscript.

González-Cao, J., García-Feal, O., Fernández-Nóvoa, D., Domínguez-Alonso, J. M., and Gómez-Gesteira, M.: Towards an automatic early warning system of flood hazards based on precipitation forecast: the case of the Miño River (NW Spain), Nat. Hazard Earth Sys., 19, 2583-2595, https://doi.org/10.5194/nhess-19-2583-2019, 2019.

Muñoz, D. F., Abbaszadeh, P., Moftakhari, H., Moradkhani, H.: Accounting for uncertainties in compound flood hazard assessment: The value of data assimilation, Coast. Eng., 171, 104057, https://doi.org/10.1016/j.coastaleng.2021.104057, 2022.

Taylor, K. E.: Summarizing multiple aspects of model performance in a single diagram, J. Geophys. Res.-Atmos., 106, 7183–7192, https://doi.org/10.1029/2000JD900719, 2001.

Wijayarathne, D. B., Coulibaly, P.: Identification of hydrological models for operational flood forecasting in St. John's, Newfoundland, Canada, J. Hydrol. Regional Studies, 27, 100646, https://doi.org/10.1016/j.ejrh.2019.100646, 2020.

- *Section 4: several parts in the discussion section should clearly be moved to the introduction or methods section because they refer to the conduct of experiments/simulation rather than the presentation and discussion of the results. Large parts thereof also contain various cited references which is typically something earlier in the manuscript. The mentioned parts are e.g.: 176-178, Section 4.2, Section 4.3 (incl subsections) until line 282; l 305-324.*

  - We agree with the reviewer in what concerns several of these suggestions. Thus, the information provided in lines 176-178 in the previous version of the manuscript has now been placed in "Data, Models and Methods" section (subsection "3.3 Digital Elevation Models").
  - The information presented in Section 4.3 in the previous version of the manuscript, including the equations related to dam operating strategies, has now been placed in "Data, Models and Methods" section.
  - However, most of the information provided in Section 4.2 in the previous version of the manuscript has been maintained in "Results and Discussion" section. In this sense, we consider that this section provides important results because it presents the numerical reconstruction of the 1979 event, even validating more in depth the capability of the model to reproduce floods in lower Tagus by comparing with some information related to the event. The results obtained from simulation are continually compared and discussed with this related information. Therefore, we consider that if some of this

**information is transferred to other sections then, this section would lose clarity. Therefore, we consider that this information should be kept in this section to facilitate the reading and understanding. However, if the reviewer considers that some parts still need to be moved to other sections we will make the changes.**

**In summary, the authors consider that with the information transferred to other sections, as recommended by the reviewer, the new version of the manuscript presents a clearer structure.**

- *L. 295ff: clearly state in the methods section that the proposed OS shall be applied to these five selected flood events*

**Done.**

- *L. 325-329: Here, the uncertainty associated to anticipate the expected volume for the coming days is mentioned. Why was this topic not addressed in an uncertainty analysis?*

**This topic was not addressed because it would imply a complete hydrological procedure along all the years, including the respective atmospheric forecasts, which are not available for the entire period under scope. In addition, the uncertainty depends not only on the precipitation forecasts used, but also on the availability of in situ measurements, and the performance of the hydrological model applied, among others. To tackle in depth all these procedures would imply a complete separate study, being well outside the scope of the present study. However, it is important to mention that currently, new approaches based on the analysis and forecast of atmospheric structures that transport large amounts of moisture (such as atmospheric rivers), which are responsible for most extreme and large intense precipitation events provide additionally predictability potential for extreme precipitation, especially for the western Iberian Peninsula (Ramos et al., 2015; 2020). However, we agree with the reviewer that further analysis focused on the uncertainty associated to these issues should be developed in future works. This information, included the aforementioned caveats, was added in the new version of the manuscript.**

Ramos, A. M., Trigo, R. M., Liberato, M. L. R., and Tome, R.: Daily precipitation extreme events in the Iberian Peninsula and its association with Atmospheric Rivers, J. Hydrometeorol., 16, 579-597, https://doi.org/10.1175/JHM-D-14-0103.1, 2015.

Ramos, A. M., Sousa, P. M., Dutra, E., and Trigo, R. M.: Predictive skill for atmospheric rivers in the western Iberian Peninsula, Nat. Hazards Earth Syst. Sci., 20, 877-888, https://doi.org/10.5194/nhess-20-877-2020, 2020.

- *L 370: maximum water velocity – if this is an important aspect, state this already in the methods or objectives section to highlight that it will be assessed in the article. Currently, it is hidden as a side note at the end of the results.*

**These important aspects to evaluate in floods, including the maximum water depth and the maximum water velocity, are now well commented in the methods section in the new version of the manuscript.**

- *Section 5: This is in large parts of a summary and no conclusion section. As mentioned above: a specific "Objectives" section might help here to motivate drawn conclusions, e.g. could the stated goals be met? Why, or why not? Are the routes for improvement?*

**Following the reviewer's suggestion, we renamed this section as "Summary and Conclusions". In addition, we rewrote this section and added information following reviewer's comments (moreover, we also added a specific section: "Motivation", as mentioned above).**

*"6 Summary and Conclusions*

[revised manuscript text omitted]

- *L. 398-405: Acknowledging caveats is an important scientific discussion but should be part of the "results and discussing" section and not of the "conclusions."*

**We agree with the reviewer. In the revised manuscript this information was placed in previous sections.**

- *L 415: In the early part of the manuscript, it was mentioned that bad communication between Portuguese and Spanish authorities exacerbated the flood impacts in 1979. Did the EU help here or is there improved bilateral operation? Please elaborate.*

**In the 1979 flood event there was a lack of communication between the different authorities that controlled the different dams, coupled with poor dam operations. This provoked that when flow peaks arrived, dams controlling Tagus flow were virtually full therefore hampering the capacity to exert sufficient control on the peak river flow. Both Portugal and Spain were not part of the EU in 1979 as both countries would enter only in 1986. Currently, a bilateral protocol is established to**

**improve the management of dams (Albufeira agreement, in 2000). In the new version of the manuscript we clarified these issues.**

Escartín, C.M.: The Agreement between Spain and Portugal for the Sustainable Development of the Shared River Basins. International Conference of Basin Organizations, Madrid, Spain, 4-6 November, 2002.

*Tables and Figures*

- *Table 2: Specify caption, e.g. which peak flow – the incoming flood wave?*

**Done (see new caption of Table 2):**

"**Table 2.** Hydrologic characteristic of most extreme flood events under different dam configurations. NR is referred to the natural regime (no dam), OS1 is referred to the operation strategy presented in equation (4), and OS2 is referred to the operation strategy focused on extreme events, presented in equation (5). Flood days are referred to the number of days exceeding the flood threshold during each considered event. Peak flow refers to the real maximum daily inflow in the case of the natural regime, whereas it is referred to the maximum daily outflow from the dam under the different operation strategies applied. Percentages are referred to the differences with respect to the worst scenario (natural regime), which is assigned a percentage of 100 %."

- *Table 2: add percentage reductions as was done in Table 3*
  **Done.**

- *Table 3: very nice overview with absolute and relative reductions!*

  **Thanks.**

- *Figure 2 caption: please add ", respectively" to the end of the sentence*
  **Done.**

- *Figure 6: please add x-axis tick labels, i.e. numbers. And specify number of day since when?*
  **Done.**

**Language**

- *Overall, the manuscript is well-written. Please read over it again to e.g. fill in missing articles (l. 20: [the] Iber+ model; l. 32: [The] Iberian Peninsula;…) or to rephrase very long sentences (e.g. l. 336-339)*

**A review of the writing of the entire manuscript was done following reviewer' suggestion.**

- *Some rewording suggestions: l. 32: "important" à "intense"; l.49-50: "that can play an important rule in" à "for"; l. 159 "free" à "freely"*

**Done.**

*References*

- *Complete*

**Thanks.**

---

## Author Comment (AC2)

First of all, the authors want to thank the referee for the work and time devoted to review the manuscript. We know that all comments will serve to improve the quality and understanding of the work and we hope we have properly answered all the suggestions.

Reviewer #2:

*The paper entitled " How to mitigate flood events similar to the 1979 catastrophic floods in lower Tagus" is well written, however, the paper is required major revision as the scientific part is missing:*

*1. The novelty part of the MS is missing in the MS. Please mention.*

**Following reviewer' suggestion, and also in accordance with the comments raised by Reviewer 1, we added a new section entitled "Motivation", in which we expose not only the motivation of the study itself, but also the novelty provided. Mainly, we commented that, on the one hand, the development of the work provides new knowledge to better understand the floods in this area since there are a scarcity of studies that address the floods on lower Tagus valley from a hydrodynamic point of view. In addition, the model validation carried out also allows providing adequate tools that can serve as a basis to perform future studies in this area. On the other hand, the proposal of dam operating strategies to take advantage of existing dams to mitigate floods in lower Tagus valley, also provide new insights since there are no previous studies that analyze this issue. Additionally, the dam operating strategies proposed can serve as a basis for future studies that even improve and optimize this proposal.**

**"2 Motivation**

The main motivations driving this study are, on the one hand, to improve the knowledge and understanding of flood development in lower Tagus valley, an area especially vulnerable to these events. In this sense, one of the main motivations for carrying out this analysis was the scarcity of studies addressing this issue, especially from a hydrodynamic point of view. For that, different freely available products were tested in order to provide the most accurate tools that can serve as a basis for future studies focused on addressing different aspects related to flooding in lower Tagus valley. On the other hand, the study also intends to provide different strategies to mitigate floods in lower Tagus valley but taking advantage of existing infrastructures, in particular, the dams. To the best of our knowledge, there are no previous studies that have developed this type of strategies for the area under scope, so the strategies presented in this work could represent an important advance in this field. This proposal will allow to provide an affordable new approach to flood mitigation compared to the implementation of additional structural measures that have to be built. For that, dam operating strategies will be proposed and tested in the most important Tagus dam. The benefits provided by the dam strategies proposed in relation to flood mitigation, will be also evaluated. This will also serve as a basis for developing future studies focused on optimizing dam strategies or even interconnecting the strategies of different dams of the Tagus basin to improve the flood mitigation."

*2. The authors have mentioned the 2D hydrodynamic model (Iber+ numerical model). Is this model is open source? Please specify.*

**The model is freely available for download from its official website (https://iberaula.es), as we stated in the manuscript, but the code is not open source. The code in only accessible for the collaborators on its development, which are specified in the web page. This information was added in the "Code and data availability" section. In the text it was also specified that what is freely available is the executable version of the model.**

*3. Please mention the comparative analysis of simulated model with Iber+ numerical model to check the accuracy of model.*

**The comparison analysis to check the accuracy of the model was better mentioned and explained in the new version of the manuscript.**

*4. The statistical analysis is missing in the MS.*

**The performance of the model to simulate floods was evaluated through the statistical analysis provided by the Taylor Diagrams. This was better explained in the new version of the manuscript, where the Taylor method is described, and the statistical results obtained were better presented and discussed. In addition, following the reviewer's comment, a more detailed statistical analysis of the comparison between the DEMs under scope and the original elevation data, was performed (Table S2 of Supplementary Material). In particular, several statistical indicators were calculated to assess the differences between the leveling benchmark altitudes and the corresponding pixel values in each DEM. These indicators include the Mean Absolute Error (MAE), which is calculated by determining the average of the absolute differences between the DEM and the benchmarks. Additionally, the Standard Deviation (SD) was computed to measure the spread of the differences between the DEM and the benchmarks. The Root Mean Squared Error (RMSE) was also computed by taking the square root of the average of the squared differences between the DEM and the benchmarks. Moreover, the Mean Error (ME) was computed as a measure of the bias between the DEM and the benchmarks, which is determined by averaging those differences. A positive ME indicates that the DEM is overestimating the elevation, while a negative ME indicates that the DEM is underestimating the elevation. This information, which is summarized in Table S2, was added to the new version of the manuscript, which increases the robustness to the validation performed.**

"In this context, and considering the wide range of available DEMs it was felt necessary to evaluate the suitability of different freely available DEMs to adequately represent floods in the lower Tagus valley. To achieve this goal, one of the most important flood events occurred in that area on February 1979, was simulated and analyzed for different DEMs in order to test which one is most appropriate for the area under

scope. As was mentioned above, four DEMs were tested, namely ESRI, ASTER, SRTM and Copernicus DEM (Karlsson and Arnberg, 2011; Wang et al., 2012; Garrote, 2022).

In general terms, the results obtained with Copernicus, SRTM and ASTER DEMs clearly indicate better performance for simulating floods in lower Tagus valley with respect to ESRI DEM, which provides worse results in all the statistics analyzed (Figure 4). Especially highlight the results obtained with Copernicus DEM, which are clearly the closest to the reference data, indicating that Copernicus DEM presents the best accuracy, i.e. the best capability to address floods in the area under scope. In particular, it presents a high correlation with the measured data, above 0.99, with a normalized standard deviation close to 1 and the lowest RMSD ($< 0.1$). The SRTM DEM also presents a correlation above 0.99, although the normalized standard deviation (1.11) and the RMSD (0.17) are worse than those obtained with Copernicus DEM. ASTER DEM presents statistics slightly worse than SRTM DEM. In addition, the original elevation data from these DEMs were also compared with the official altimetric values by calculating several statistical indicators to evaluate the associated error and deviation (see Table S2 in the Supplementary Material). Copernicus DEM is also corroborated as the most accurate, presenting the lowest values in all the analyzed statistics, followed again by the SRTM DEM (see the detailed analysis provided in the Supplementary Material). Recent studies comparing the accuracy of different DEMs along the European continent (Guth and Geoffroy, 2021) and in other parts of the world (Garrote, 2022), also confirm the higher precision of Copernicus DEM in comparison with other global products.

This confirms that Copernicus DEM, coupled with the Iber+ model, are capable of the adequate reproduction, at large-scale, of the flood events in the lower Tagus. In fact, the statistical parameters analyzed by means of the Taylor diagrams corroborate not only the better performance compared to the other DEMs analyzed, but also the accurate representation of the reference flood data. Therefore, Copernicus DEM was selected for the remaining of the analysis."

[Figure]

**Figure 4.** Taylor diagram of the water elevation obtained with Iber+ using the field data as reference. E, A, S and C indicate the Iber+ data obtained using the ESRI, ASTER, SRTM and Copernicus Digital Elevation Models, respectively.

| STATISTICAL INDICATOR \ DEM | ESRI | ASTER | SRTM | Copernicus |
|---|---|---|---|---|
| MAE (m) | 3.56 | 4.74 | 3.10 | 2.12 |
| SD (m) | 4.71 | 4.90 | 3.94 | 3.81 |
| RMSE (m) | 4.81 | 5.91 | 4.42 | 3.81 |
| ME (m) | -1.00 | 3.30 | 2.01 | 0.17 |

**Table S2.** Statistical analysis of the altitude difference between leveling benchmarks and analyzed DEMs.

**5. Please do the sensitivity analysis of the model.**

**Following a similar suggestion provided by both reviewers, a sensitive analysis was performed to analyze the effectiveness of the proposed strategies to mitigate floods in lower Tagus valley under different river flow scenarios, taking into account that these strategies suppose the main tool developed to address flood mitigation. For that, the original series of river flow was randomly perturbed allowing a deviation of ± 25 %, that is, each real daily value of river flow has been allowed a random variation of ± 25 %. In this sense, as many perturbed series as the original number of data were generated (> 17000) to add more robustness to the evaluation. The average number of floods generated by the river flow of the perturbed series as well as the associated standard deviation were presented in a new table. The respective average number of floods resulting from applying the dam operating strategies proposed to the perturbed series, as well as the associated standard deviation, were also evaluated. The efficiency of both proposed strategies was clearly maintained in terms of reducing the total number of floods. In addition, the effectiveness of OS2 to mitigate the most extreme floods was also confirmed. The results obtained corroborate the robustness of dam operating strategies proposed under different scenarios of river flow. This information was added in the new version of the manuscript.**

| Parameter | Natural Regime | Operation Strategy OS1 | Operation Strategy OS2 |
|---|---|---|---|
| *Days > 1000 m³s⁻¹* | 453.97 ± 87.63 | 77.64 ± 3.34 | 89.05 ± 3.57 |
| *Days > 3000 m³s⁻¹* | 37.98 ± 3.16 | 16.13 ± 2.05 | 14.42 ± 2.28 |
| *Days > 5000 m³s⁻¹* | 6.98 ± 1.61 | 3. 65 ± 0.88 | 1.66 ± 0.98 |

**Table.** The original series of real inflow at Alcántara location was perturbed by applying a random deviation of ± 25% to the daily river flow values. Thus, several random perturbed series equal to the total number of days from the original series (> 17000 days) were created. Then, the mean number of days (and the corresponding standard deviation) exceeding different critical outflows at Alcántara location, were calculated for the perturbed river flow series, considering a natural regime and also, applying the operation strategies OS1 and OS2.

*6. Kindly, separate the discussion section and mention the limitation and recommendation of the study.*

**Following similar suggestions by both reviewers, we have restructured the manuscript to a certain extent. In this sense, and following also the comments of reviewer 1, some parts of the "Results and Discussion" section, more related to conduct experiments or simulations, as well as some statements or information that are not specifically a result, have now been placed in previous sections ("Introduction" and "Data, Models and Methods" sections).**

**The limitations and recommendations of the study were also clarified in the revised version of the manuscript. The limitations and caveats are now exposed in the corresponding sections, while the recommendations were placed both in the "Motivation" section and also in the "Summary and Conclusions" section.**

**We consider that the new version of the manuscript now presents a clearer structure. However, if the reviewer considers that more changes are needed, we will make the proposed additional changes.**

---

## Referee Report (RR1)

**nhess-2022-243**

Title: How to mitigate flood events similar to the 1979 catastrophic floods in lower Tagus

**Reviewer Comments:**

The MS can be considered for the publication.

---

## Referee Report (RR2)

**Summary**

The research work presented in the paper « How to mitigate flood events similar to the 1979 catastrophic floods in lower Tagus » presents a comprehensive study of flood mitigation strategies with dam to reduce flood impacts for the Iberian Peninsula.

The Iber+ numerical model was used with different DEMs: i) to select the most relevant DEM for the study area, ii) to model the 1979 Tagus River floods, and iii) to propose a management strategy for the Alcántara dam.

**Evaluation and recommandations**

The manuscript is overall well-written and contains many relevant bibliographic references. The chosen structure for the paper is coherent, with a good description of the data and models used.

It should also be noted that the authors have taken into account the comments of previous reviewers very well. There are still a few minor points of detail remaining, and I recommend a minor revision for this new manuscript.

The following comments aim to guide the finalization of the paper.

**Comments**

l.147 – please define SCS-CN

Section 4 – Proposal for a reorganization of this section; the title is not consistent as no methods are clearly presented here. It might be preferable to add a subsection 4.4 « method » for l.175-193.

Section 4 – In general, regarding the method: a few sentences could be added about the use of current DEMs to simulate an event from 30 years ago, with the possibility of relying on historical data (photos, etc.) to validate the DEM used.

l.194-199 – Repetition with l.56-57

l.331 – Introducing abbreviations such as RMSD, for example (not defined and used in line 186).

Section 6 – This section appears more like a summary of the work, lacking critical analysis, identified limitations, and whether the stated objectives were achieved (which are not mentioned elsewhere).

Figure 1 – numbering the figures would aid in understanding the legend. The main affected villages could be directly presented on the main figure.

General comments:

1. It would be helpful to add hyperlinks to navigate to figures and references.

2. Replace "Tagus river" with "Tagus River."

3. Standardize abbreviations: sometimes DEM, sometimes DEMs.

4. Almost all maps presented without scale and orientation.

---

## Referee Report (RR3)

**Summary**

The research work presented in this paper "How to mitigate flood events similar to the 1979 catastrophic floods in lower Tagus" describes a comprehensive study of flood mitigation strategies with dam to reduce flood impacts for the Iberian Peninsula.

The Iber+ numerical model was used with different DEMs: i) to select the most relevant DEM for the study area, ii) to model the 1979 Tagus River floods, and iii) to propose a management strategy for the Alcántara dam.

**Evaluation and recommendations**

I note that the recommendations made at the last review have been followed.

Section 6 – I still think section 6 is a little light on analysis. Simple elements based on photos and high-water marks from the period could have been used to justify and validate the DEM used. The added section on the use of more precise DEMs does not necessarily seem relevant, as this does not play a part in validating the model.

---

## Author Response (AR2)

**The authors would like to thank the referees for the work and time devoted to review the manuscript. The remaining comments and suggestions raised by all reviewers were useful to improve the overall quality and understanding of the work.**

**Lines are referred to the marked-up revised version of the manuscript.**

**Reviewer #1:**

*The MS can be considered for the publication.*

**The authors would like to thank the referee for reviewing the article and we sincerely appreciate the positive feedback.**

**Reviewer #2:**

**Summary**

*The research work presented in the paper « How to mitigate flood events similar to the 1979 catastrophic floods in lower Tagus » presents a comprehensive study of flood mitigation strategies with dam to reduce flood impacts for the Iberian Peninsula.*

*The Iber+ numerical model was used with different DEMs: i) to select the most relevant DEM for the study area, ii) to model the 1979 Tagus River floods, and iii) to propose a management strategy for the Alcántara dam.*

*Evaluation and recommendations*

*The manuscript is overall well-written and contains many relevant bibliographic references. The chosen structure for the paper is coherent, with a good description of the data and models used.*

*It should also be noted that the authors have taken into account the comments of previous reviewers very well. There are still a few minor points of detail remaining, and I recommend a minor revision for this new manuscript.*

*The following comments aim to guide the finalization of the paper.*

*Comments*

*l.147 – please define SCS-CN*

**Done.**

*Section 4 – Proposal for a reorganization of this section; the title is not consistent as no methods are clearly presented here. It might be preferable to add a subsection 4.4 « method » for l.175- 193.*

**This section was reorganized following the reviewer's comment. Thus, a new subsection *"4.4 Validation method of the hydraulic model coupled with different DEMs"* was added containing the information previously presented in lines 175-193 of the preceding version of the manuscript.**

*Section 4 – In general, regarding the method: a few sentences could be added about the use of current DEMs to simulate an event from 30 years ago, with the possibility of relying on historical data (photos, etc.) to validate the DEM used.*

**Some sentences have been added about using current DEMs to simulate the past events under scope (lines 183-191). In this new paragraph we discuss the macroscopic scope of the study, the lack of precise and well distributed older terrain data, and the need to establish a common framework to compare and evaluate the differences in flood impact under the mitigation strategies proposed.**

*l.194-199 – Repetition with l.56-57*

**Sentences were rewritten to avoid repetitions.**

*l.331 – Introducing abbreviations such as RMSD, for example (not defined and used in line 186).*

**This was revised and in this new version of the manuscript all abbreviations used were defined the first time they appear.**

*Section 6 – This section appears more like a summary of the work, lacking critical analysis, identified limitations, and whether the stated objectives were achieved (which are not mentioned elsewhere).*

**Some parts of this section were rewritten according to reviewer's suggestion.**

*Figure 1 – numbering the figures would aid in understanding the legend. The main affected villages could be directly presented on the main figure.*

**Done.**

*General comments:*

*1. It would be helpful to add hyperlinks to navigate to figures and references.*

**The authors also think this would be useful, but we would like to stress that this typeset choice depends on the preferences of the journal and that this should be done by the copy-editing department of the journal in case the paper is accepted for publication.**

*2. Replace "Tagus river" with "Tagus River."*

**Done.**

*3. Standardize abbreviations: sometimes DEM, sometimes DEMs.*

**DEM is used throughout the text when referring to a single Digital Elevation Model, and the plural, DEMs, is used when referring to multiple Digital Elevation Models. However, when reviewing the text in depth, we realized that in some places the abbreviation was not correct, and it was corrected throughout the entire manuscript (i.e. the term DEM is used when referring to a singular Digital Elevation Model and DEMs when referring to a several Digital Elevation Models).**

*4. Almost all maps presented without scale and orientation.*

**Scale and orientation were added in all maps.**

Reviewer #3

*The paper is relevant to the scope of the journal. It is well written. The results are well elaborated and cited with suitable tables and figures. The lesson learned from the past event and improved the flood mitigation is important in strategic planning. Present case, author(s) has applied for flood mitigation planning at Tagus Valley. The operation of Alcantara dam and analysis has been performed using hydraulic modeling. Useful decision-making results and conclusion has been derived to reinforce the decision-making system.*

*Some useful suggestions/ comments to improve the quality of the paper:*

*1) Author has utilized Iber+ numerical model to simulate the flow in 2D. The daily mean river flow or flood hydrograph has considered for upstream boundary, however, How the downstream boundary considered was little vague? Pl. clarify in section 4.2.*

**We agree with the reviewer. In the new version of the manuscript the downstream boundary conditions were better explained in the section devoted to describing the hydraulic model (Section 4.2, Hydraulic model; lines 140-142).**

*2) The author has utilized different land use data from CORINE Land cover data, However, the resolution of the land use file and classes of land use considered for modeling were missing. Furthermore, land use roughness plays a major role on arrival time and velocity of flow, the roughness vales of each land use for flood plain simulation were missing?*

**Following reviewer's suggestions, a new figure was added specifying the land uses and the associated manning coefficients (new Figure 2). In addition, more information, such as the resolution of the land use database, was added in the text (lines 148-150).**

*3) The 2D flow grid is an important simulation parameter that decides the simulation time, however, the 2D flood cell and interval the model was simulated are not presented by the applied model.*

**This important information was added in the new version of the text (lines 151-154). It was specified that the domain was discretized in a mesh of unstructured triangles with average side lengths that varied from 25 to 100m, surpassing 6M of total elements, and that the average computational time step was variable and was calculated following the Courant-Friedrichs-Levy (CFL) condition.**

*4) The downstream inline structure i.e. bridges, bridges piers affect the 2D flow, and how this feature was emitted for building the model. If it is so, elaborate the limitation in the discussion part of the paper.*

**Due to the macroscopic view of this study, where the main purpose was to analyse the macroscopic response of the entire Tagus valley area, the downstream inline structures were not considered in this analysis, since, additionally, they would deserve an individual and detailed analysis, and this is beyond the scope of the present article. The limitations and recommendations were also stablished and discussed in the paper, remarking the need to also consider these structures to perform local flood analyses where they are involved, and especially when high resolution information is available. This information was added in the new version of the manuscript. (lines 155-160).**

*5) The soil Aggradational and degradational features and its related time series analysis for flood inundation were missing in present modeling.*

**The aggradational and deggradational features were not considered. This was clarified in the new version of the manuscript (lines 155-157).**

*6) Author(S) has claimed the model accuracy and outcome depends on DEM resolution, however, the flood flow simulation depends on various parameters i.e. 2D flood cell size, DEM resolution, courant number, river and flood plain roughness, etc. Therefore, the accuracy of model outcome claimed only by DEM resolution is difficult to judge. Although, DEM resolution would be one of the parameters to improve the accuracy of the model and in this case, it would be justifiable to compare the model with different DEM resolutions.*

**We agree with the reviewer that the model accuracy depends on more parameters than the DEM resolution, although for the present case this may be one of the most important factors. In the new version of the manuscript, we discuss these issues and also comment on the associated limitations (lines 544-552).**

*Some useful literature for author(s) consideration:*

*Shah Z, Saraswat A., Samal D. and Patel DP (2022), "A Single Interface for Rainfall-Runoff Simulation and Flood Assessment – A Case of New Capability of HEC-RAS for*

*Flood Assessment and Management", Arabian Journal of Geosciences, Springer. 15, 1526(2022) https://doi.org/10.1007/s12517-022-10721-2.;*

*Pathan, A.I., Agnihotri P.G , and Patel, D.P. (2022) " Integrated approach of AHP and TOPSIS (MCDM) techniques and GIS for dam site suitability mapping : a case study of Navsari City, Gujarat, India. Environmental Earth Science, Springer. (2022) 81:443. https://doi.org/10.1007/s12665-022-10568-6;*

*Prieto C., Patel DP, and Han D (2020), "Preface: Advances in flood risk assessment and management", Nat. Hazards Earth Syst. Sci., 20, 1045–1048, https://doi.org/10.5194/nhess-20-1045-2020;*

*Patel DP, Jorge AR, Srivastava PK, Michaela B. and Han D. (2017). "Assessment of flood inundation mapping of Surat city by coupled 1D/2D hydrodynamic modeling- A case application of the new HEC-RAS 5". Natural Hazards, Springer, 89(1): 93-130. https://doi.org/10.1007/s11069-017-2956-6*

**Thanks for this valuable information. Authors have included these references and the related information, in the new version of the manuscript.**

*Minor comments:*

*1) The land use land cover map is missing.*

**A new figure with the land uses and the associated roughness manning values was added in the new version of the manuscript (new Figure 2).**

*2) The table and figure caption is too long. It usually has 1 or 2 line short title with required core information. Therefore, it would suggest to concise the table and figure caption.*

**Some of the table and figure captions were simplified according to reviewer's suggestion.**

*3) Fig. 5 scale bar is missing*

**Scale bar was added.**

*4) Fig. 6 north arrow, legend and scale bar is missing*

**North arrow, legend and scale bar were added.**

*5) Fig. 8 improves the resolution upto 400 dpi.*

**Done.**

*It is suggested to incorporate and justify the raised comments, and submit it in the revise version for further consideration.*

**Authors hope to have properly answered all the comments and to have added the changes and corrections proposed.**